# Adaptive Online Convex Optimization via Sparse-Low-Rank Gradient Decomposition

## Abstract

Adaptive gradient methods for online convex optimization often require the practitioner to choose in advance between diagonal preconditioning, which exploits coordinate sparsity, and full-matrix preconditioning, which exploits low-rank gradient geometry. We propose SLR-FTRL, a structure-adaptive model-selection method that decomposes the gradient preconditioner into sparse and low-rank components, runs two parallel FTRL sub-algorithms with structurally matched preconditioners, and combines their outputs through a coin-betting meta-algorithm that learns which preconditioner, or mixture of preconditioners, performs best in hindsight. Our main guarantee is a best-mixture regret bound of the form $\min_{\alpha^*}\{\alpha^* R_T^L + (1 - \alpha^*)R_T^S\} + \widetilde{O}(\sqrt{T})$. Thus, the theoretical advantage of SLR-FTRL is a model-selection/adaptive-interpolation guarantee rather than a claim that a single joint sparse-low-rank preconditioner always improves both base bounds simultaneously: the algorithm avoids committing to a mismatched geometry and recovers the classical diagonal and full-matrix AdaGrad guarantees as special cases when one structure is absent, with all cross-contamination and preconditioner-lag corrections explicitly retained. We further establish per-term lower bounds showing that the structural dependence on $\sqrt{r \cdot \sigma_1(\mathbf{G}_T^L)}$ and $\sqrt{s \cdot \|\mathbf{G}_T^S\|_\infty}$ is individually tight up to constants. Experiments on online regression with structured gradients confirm the theoretical predictions, demonstrating sublinear regret, pure-case recovery, dimension independence, and graceful degradation under decomposition noise.

## 1 Introduction

The effectiveness of adaptive gradient methods in modern machine learning, from sparse feature spaces in online advertising to the low-rank structure of neural network loss landscapes, depends on how well they exploit the geometry of gradient sequences. Online convex optimization (OCO) provides the theoretical foundation for these methods, formalizing a sequential decision-making setting in which a learner repeatedly selects a point $w_t$ from a convex set $\mathcal{W} \subseteq \mathbb{R}^d$, observes a convex loss function $f_t$, and aims to minimize the cumulative regret $\sum_{t=1}^{T} f_t(w_t) - \min_{w^* \in \mathcal{W}} \sum_{t=1}^{T} f_t(w^*)$ (Shalev-Shwartz, 2025; Hazan, 2016). In modern high-dimensional applications, such as training large language models, optimizing sparse feature interactions in recommender systems, or continual fine-tuning of foundation models, gradient sequences often exhibit a mixture of structural properties: only a small subset of coordinates may receive nonzero updates at any given round (sparsity), while the active gradients may concentrate along a low-dimensional subspace (low-rank structure). Existing adaptive methods are designed to exploit only one of these structures at a time, and the question of how to adaptively select or interpolate between sparse and low-rank preconditioning within a single algorithm, without knowing in advance which structure will dominate, has remained open.

The development of geometry-adaptive online learning has progressed through several phases, each targeting a different aspect of gradient structure. Online gradient descent (OGD) (Zinkevich, 2003) established the foundational $O(\sqrt{T})$ regret rate for general convex losses but employs a scalar learning rate that treats all gradient coordinates uniformly, ignoring any structure in the gradient sequence. The Online Newton

Step (Hazan et al., 2007) achieved $O(d \log T)$ regret for the restricted class of exp-concave losses by leveraging second-order curvature information, yet this strong assumption is rarely satisfied in practice. The introduction of AdaGrad (Duchi et al., 2011; McMahan & Streeter, 2010) showed that adaptive preconditioning based on accumulated gradient second moments can yield data-dependent bounds that are never worse than, and often dramatically better than, those of non-adaptive methods. This insight gave rise to two fundamentally different preconditioner designs. The *diagonal* variant constructs per-coordinate learning rates and achieves regret proportional to $\sum_{j=1}^{d} (\sum_{t=1}^{T} g_{t,j}^2)^{1/2}$, which reduces to $O(s\sqrt{G_\infty})$ when only $s \ll d$ coordinates are ever active (where $G_\infty = \max_j \sum_t g_{t,j}^2$), which is much tighter in sparse regimes since inactive coordinates contribute nothing to the bound. However, diagonal preconditioning treats each coordinate independently and cannot capture correlations among gradient coordinates. The *full-matrix* variant uses the matrix square root $(\sum_t g_t g_t^\top)^{1/2}$ as a preconditioner, yielding regret proportional to $\mathrm{tr}(\mathbf{G}_T^{1/2})$, which can be as small as $O(r \cdot \sigma_1(\mathbf{G}_T)^{1/2})$ when gradients lie in a fixed $r$-dimensional subspace, but at $O(d^2)$ per-round cost and with no ability to exploit coordinate-wise sparsity. Subsequent work has built on these ideas: Kronecker-factored approximations have scaled full-matrix preconditioning to large neural networks (Gupta et al., 2018); coin-betting reductions have removed the need for learning rate tuning (Orabona & Pál, 2016; Cutkosky & Orabona, 2018); and recent analyses have further improved the understanding of adaptive methods under relaxed assumptions (Défossez et al., 2020; Jacobsen & Cutkosky, 2022; Mhammedi & Koolen, 2020). Yet a basic limitation remains: *every existing method must commit to a single type of structural assumption*, either sparsity or low-rank, so choosing the wrong preconditioner can yield the worse of the two structural guarantees. The goal of this work is to remove this manual choice by competing with the better structural preconditioner, or the better hindsight mixture of the two, rather than to prove a universally smaller joint sparse-low-rank leading term.

In this paper, we propose **SLR-FTRL** (Sparse-Low-Rank Follow-The-Regularized-Leader), a structure-adaptive model-selection algorithm that addresses this limitation by maintaining sparse and low-rank preconditioners in parallel and learning their mixture online. At each round, the observed gradient $g_t$ is decomposed into a sparse estimate $\hat{S}_t$ and a low-rank estimate $\hat{L}_t$ via an online robust PCA oracle (Feng et al., 2013; Goes et al., 2014). Two parallel FTRL sub-algorithms are maintained: one employing a diagonal preconditioner $D_t = \mathrm{diag}((\delta + \sum_{\tau \le t} \hat{S}_{\tau,j}^2)^{1/2})$ built from the sparse components, and another employing a full-matrix preconditioner $M_t = (\delta I + \sum_{\tau \le t} \hat{L}_\tau \hat{L}_\tau^\top)^{1/2}$ built from the low-rank components. A key design principle is that both sub-algorithms receive the *full gradient* $g_t$ as their loss signal (not the decomposed components), so both compete against the same comparator $w^*$ and the decomposition error affects only preconditioner quality, not the loss sequence. A coin-betting meta-algorithm (Orabona & Pál, 2016) produces the convex combination $w_t = \alpha_t w_t^L + (1 - \alpha_t) w_t^S$, automatically competing with the best hindsight mixing coefficient $\alpha^*$ between diagonal and full-matrix preconditioning; this is the model-selection sense in which SLR-FTRL adapts to the structural regime. Table 1 summarizes how SLR-FTRL compares to existing methods under the two canonical structural assumptions. Under pure-structure special cases ($r = 0$ or $s = 0$), cross-contamination terms vanish and SLR-FTRL recovers the classical diagonal and full-matrix AdaGrad bounds up to lower-order corrections; in the general mixed case, the minimization over mixing coefficients $\alpha^*$ automatically selects the more favorable structural regime. The lower bound row in Table 1, which extends classical minimax arguments (Cutkosky et al., 2023) to gradient-structured settings, confirms that the leading structural terms $\sqrt{r\sigma_1}$ and $\sqrt{sG_\infty}$ in our upper bound are individually tight up to constants.

We emphasize the precise scope of this guarantee. The upper bound is a model-selection/adaptive-interpolation guarantee: SLR-FTRL competes with the best hindsight convex combination of a low-rank-preconditioned learner and a sparse-preconditioned learner. It should not be read as a joint-exploitation guarantee that produces a new leading term strictly smaller than both $R_T^L$ and $R_T^S$ on every mixed instance. The advantage over classical single-preconditioner methods is instead that SLR-FTRL avoids committing to a potentially mismatched geometry: since the endpoints $\alpha^* = 0$ and $\alpha^* = 1$ are allowed, Theorem 1 implies $\mathrm{Regret}_T(w^*) \le \min\{R_T^L, R_T^S\} + \tilde{O}(\sqrt{T})$ under the bounded-iterate regime, and hence can be much better than the worse structural choice whenever one of the two geometries is more favorable. Designing a single preconditioner with a genuinely joint sparse-low-rank bound remains an open direction.

Our main contributions are summarized as follows:

Table 1: Comparison of online learning methods under pure structural assumptions ($\epsilon = 0$). Leading-order regret terms shown; constants and log factors suppressed. $D = \|w^*\|$, $G_\infty = \max_j \sum_t g_{t,j}^2$, $\sigma_1 = \sigma_1(\mathbf{G}_T)$. "PF" = parameter-free. "St." = structural model selection/adaptive interpolation between sparse and low-rank regimes.

| Method | $s$-sparse bound | Rank-$r$ bound | Cost | PF | St. |
|---|---|---|---|---|---|
| *Classical methods* | | | | | |
| OGD (Zinkevich, 2003) | $DG\sqrt{T}$ | $DG\sqrt{T}$ | $O(d)$ | ✗ | ✗ |
| Diag. AdaGrad (Duchi et al., 2011) | $(D^2+s)\sqrt{G_\infty}$ | $(D^2+\sqrt{dr})\sqrt{\sigma_1}$ | $O(d)$ | ✗ | ✗ |
| Full-Mat. AdaGrad (Duchi et al., 2011) | $(D^2+s)\sqrt{\sigma_1}$ | $(D^2+r)\sqrt{\sigma_1}$ | $O(d^2)$ | ✗ | ✗ |
| *Structured / parameter-free methods* | | | | | |
| Shampoo (Gupta et al., 2018) | $(D^2+s)\sqrt{\sigma_1}$ | $(D^2+r)\sqrt{\sigma_1}$ | $O(d^{4/3})$ | ✗ | ✗ |
| Coin-Betting (Orabona & Pál, 2016; Cutkosky & Orabona, 2018) | $DG\sqrt{T \ln T}$ | $DG\sqrt{T \ln T}$ | $O(d)$ | ✓ | ✗ |
| PF Mirror Desc. (Jacobsen & Cutkosky, 2022) | $DG\sqrt{T \ln T}$ | $DG\sqrt{T \ln T}$ | $O(d)$ | ✓ | ✗ |
| DoWG (Khaled et al., 2023) | $DG\sqrt{T}$ | $DG\sqrt{T}$ | $O(d)$ | ✓ | ✗ |
| **SLR-FTRL (Ours)** | $(D^2+s)\sqrt{G_\infty}+\widetilde{O}(\sqrt{T})$ | $(D^2+r)\sqrt{\sigma_1}+\widetilde{O}(\sqrt{T})$ | $O(dr^2)$ | ✓ | ✓ |

**Lower bounds** (Theorem 2): $\Omega(D\sqrt{sG_\infty})$ for $s$-sparse gradients; $\Omega(D\sqrt{r\sigma_1})$ for rank-$r$ gradients. The leading structural terms in SLR-FTRL match these lower bounds up to constants. The $\widetilde{O}(\sqrt{T})$ term for SLR-FTRL is the meta-algorithm overhead from combining the two preconditioners, under the standing assumption that the FTRL iterates are bounded by a problem-dependent constant $B_w = O(1)$ (the same regime in which the diagonal/full-matrix AdaGrad baselines themselves produce $O(1)$ iterates and adaptive preconditioning is informative). Under this assumption, the overhead is of the same $\sqrt{T}$ order as the parameter-free baselines (Coin-Betting, PF Mirror Descent, DoWG) and is strictly lower-order than the leading structural terms $(D^2 + s)\sqrt{G_\infty}$ or $(D^2 + r)\sqrt{\sigma_1}$ whenever $G_\infty$ or $\sigma_1$ scales linearly with $T$. The worst-case dependence on $B_w$ is analyzed in Remark 2 and Appendix C.

- **Algorithm.** We introduce SLR-FTRL as a structure-adaptive model-selection algorithm over sparse and low-rank preconditioners. The algorithm decomposes each gradient's preconditioner into sparse and low-rank components via an online robust PCA oracle, maintains two parallel FTRL sub-algorithms with structurally matched preconditioners, and combines their outputs through a coin-betting meta-algorithm that learns the best hindsight mixing coefficient between the two base learners.[1] A key design principle is that both sub-algorithms receive the full gradient as their loss signal, so the decomposition error affects only preconditioner quality rather than the loss sequence.

- **Upper bound.** We prove a regret bound of the form $\min_{\alpha^*}\{\alpha^* R_T^L + (1 - \alpha^*)R_T^S\} + \widetilde{O}(\sqrt{T})$, where $R_T^L$ and $R_T^S$ are the regret of the low-rank and sparse sub-algorithms with all cross-contamination and preconditioner-lag terms explicitly retained. This is explicitly a best-hindsight-mixture/model-selection guarantee: because $\alpha^* = 0$ and $\alpha^* = 1$ are feasible, SLR-FTRL competes with the better of the two structural learners up to meta-overhead, and can therefore avoid the worse guarantee produced by a mismatched preconditioner. When one structure is absent, cross-contamination terms vanish and the bound recovers the classical diagonal AdaGrad guarantee under pure sparsity and the full-matrix AdaGrad guarantee under pure low-rank, up to lower-order corrections.

- **Lower bound.** We establish per-term minimax lower bounds of $\Omega(D\sqrt{r \cdot \sigma_1(\mathbf{G}_T^L)})$ and $\Omega(D\sqrt{s \cdot \|\mathbf{G}_T^S\|_\infty})$ via separate adversarial constructions for the low-rank and sparse regimes, respectively. These matching per-term lower bounds are surfaced as a main contribution because they identify exactly which structural dependences in the two base guarantees cannot be improved in general. These results confirm that the leading structural terms in our upper bound are individually tight up to constants.

---

[1]We clarify the scope of this claim. The meta-algorithm is parameter-free in that it adapts to the optimal mixture $\alpha^*$ of the sparse and low-rank sub-learners without knowing $r$, $s$, or which structure dominates, and is also parameter-free in the comparator norm $\|w^*\|$ and step size in the standard sense of Orabona & Pál (2016); Cutkosky & Orabona (2018). The decomposition oracle (Lines 7 to 9 of Algorithm 1) still takes $r$ and $s$ as inputs to instantiate $\mathrm{Proj}_r$ and $\mathrm{Hard}_s$, and is therefore not parameter-free in the structural parameters themselves. SLR-FTRL thus achieves adaptivity to the structural regime, which of the two structures or which mixture drives the bound, rather than full parameter-freeness in $r$ and $s$. Removing the dependence on $r$ and $s$ in the decomposition step (e.g., via adaptive rank/sparsity selection) is an interesting direction for future work.

- **Empirical validation.** We empirically validate the model-selection guarantee on online regression with controlled sparse-low-rank gradients. The experiments show pure-case recovery of the specialized AdaGrad baselines, approximate dimension independence at fixed structural complexity, and graceful degradation under decomposition noise, while also quantifying the small overhead paid relative to the better specialized baseline.

## 2 Related Work

Our work connects three lines of research: adaptive preconditioning in online learning, parameter-free algorithms via coin-betting reductions, and online sparse-plus-low-rank matrix decomposition. We review each in turn, highlighting the foundations we build on and the key differences from our approach.

### 2.1 Adaptive Preconditioning in Online Learning

The Follow-The-Regularized-Leader (FTRL) framework (McMahan, 2011; Shalev-Shwartz, 2025) provides a unified perspective on online learning algorithms, in which the choice of regularizer, and in particular the preconditioner used to scale gradient updates, determines the algorithm's sensitivity to problem structure. Within this framework, the preconditioner $H_t$ in the quadratic regularizer $\frac{1}{2}\|w\|^2_{H_t}$ controls the tradeoff between expressiveness and computational cost.

The diagonal choice underlying AdaGrad (Duchi et al., 2011; McMahan & Streeter, 2010) has been widely adopted and inspired a family of practical adaptive optimizers. Adam (Kingma & Ba, 2014) augments diagonal adaptivity with exponential moving average momentum, making it the de facto optimizer for deep learning; however, Reddi et al. (2019) identified a critical flaw in the original convergence analysis and proposed AMSGrad as a corrected variant. Ward et al. (2020) established that AdaGrad stepsizes achieve sharp convergence rates even over nonconvex landscapes, extending the scope of diagonal preconditioning beyond the online convex setting. Défossez et al. (2020) later provided streamlined convergence proofs unifying the analyses of Adam and AdaGrad under a common framework. All diagonal methods share a common limitation: they cannot capture correlations among gradient coordinates. The preconditioner is coordinate-separable by design, so low-rank structure in the gradient outer-product matrix $\mathbf{G}_T$ cannot reduce the regret bound below $O(\sum_j \sqrt{G_{jj}})$.

The full-matrix variant of AdaGrad (Duchi et al., 2011) addresses this limitation by using $H_T = (\mathbf{G}_T)^{1/2}$, yielding the trace bound $\mathrm{tr}(\mathbf{G}_T^{1/2})$ that can be dramatically smaller when gradients are low-rank. The practical obstacle is the $O(d^2)$ per-round cost. Shampoo (Gupta et al., 2018) mitigates this by approximating the full preconditioner via Kronecker-product structure across tensor dimensions, making full-matrix-style preconditioning tractable for large-scale parameters. An alternative route is matrix sketching: the Frequent Directions algorithm (Ghashami et al., 2016) maintains a rank-$k$ approximation of the streaming outer-product matrix in $O(dk)$ space, offering a continuous tradeoff between the diagonal ($k = 0$) and full-matrix ($k = d$) extremes. However, sketch-based preconditioners introduce approximation error that may offset the benefit of capturing low-rank structure, and they provide no mechanism to simultaneously exploit coordinate-wise sparsity.

Unlike all of the above methods, which commit to a single preconditioner family, SLR-FTRL maintains *both* a diagonal and a full-matrix preconditioner simultaneously through independent sub-algorithms, each receiving the full gradient but exploiting different structure. The meta-algorithm then determines the optimal blend, so that the practitioner need not choose between them. The resulting theory is therefore best understood as structural model selection among preconditioner families; a single preconditioner that jointly exploits sparse and low-rank structure with a strictly improved mixed-structure leading term is left as future work.

### 2.2 Parameter-Free Online Learning and Coin-Betting Reductions

A parallel line of research has focused on removing the dependence on unknown problem parameters, such as the time horizon $T$, the gradient bound $G$, or the comparator norm $\|w^*\|$, when setting the learning rate. Classical expert aggregation strategies, including the multiplicative weights method (Freund & Schapire,

1997) and the Hedge algorithm (Cesa-Bianchi & Lugosi, 2006), require prior knowledge of the loss range or a tuned exploration parameter. De Rooij et al. (2014) provided a refined understanding of when follow-the-leader suffices versus when hedging is necessary, showing that the leader's stability properties determine the optimal strategy. Van Erven & Koolen (2016) proposed MetaGrad, which maintains a pool of candidate learning rates and aggregates their predictions to achieve regret that adapts to the best rate in hindsight, effectively obtaining quantile-type bounds without prior knowledge. Luo & Schapire (2015) developed AdaNormalHedge, which adapts to heterogeneous expert quality and achieves near-optimal aggregation bounds without any tunable parameters.

The coin-betting framework of Orabona & Pál (2016) reduces online linear optimization to the problem of sequentially wagering on a binary outcome. The Krichevsky–Trofimov (KT) estimator, originally designed for universal data compression, yields a wealth process whose logarithmic growth rate is controlled by the regret of the underlying online learning problem. This reduction produces algorithms whose regret automatically adapts to $\|w^*\|$ without requiring it as input. Cutkosky & Orabona (2018) generalized this framework to arbitrary Banach spaces, providing modular black-box reductions that convert any coin-betting strategy into a parameter-free online learner. More recent extensions include parameter-free mirror descent with Bregman divergences (Jacobsen & Cutkosky, 2022) and simultaneous adaptivity to both the Lipschitz constant and comparator norm (Mhammedi & Koolen, 2020).

Our use of coin-betting differs from prior work in an important way. Existing applications of the coin-betting reduction focus on eliminating *scalar* hyperparameters, typically the learning rate or a norm bound, within a single algorithm. In SLR-FTRL, the coin-betting meta-algorithm instead solves a *structural model selection* problem: it treats the sparse and low-rank sub-algorithms as two experts with complementary inductive biases and learns the optimal mixing coefficient $\alpha^*$ online. This application of coin-betting to combine structurally *different* preconditioner families, rather than to tune a scalar parameter within a single preconditioner, is, to our knowledge, new. Accordingly, the claimed advantage is the ability to track the better structural model in hindsight, not a claim that the upper bound is a new joint sparse-low-rank rate smaller than both base-learner bounds on every sequence.

### 2.3 Sparse-Plus-Low-Rank Decomposition

The decomposition of a matrix $M$ into the sum $M = L + S$ of a low-rank component $L$ and a sparse component $S$ is well studied in high-dimensional statistics and signal processing. The foundational work of Candès et al. (2011) on Principal Component Pursuit demonstrated that, under appropriate incoherence conditions, exact recovery of both components from their superposition is possible via convex optimization, specifically minimizing $\|L\|_* + \lambda \|S\|_1$. Chandrasekaran et al. (2011) characterized the algebraic conditions, termed rank-sparsity incoherence, under which the decomposition is identifiable, providing necessary and sufficient conditions for the convex relaxation to succeed. These batch results assume access to the full data matrix and solve an offline optimization problem.

The extension to online and streaming settings introduces additional challenges, as the algorithm must process observations sequentially and cannot revisit past data. Feng et al. (2013) proposed an online robust PCA method based on stochastic optimization that provably recovers the low-rank subspace under column-sparse outlier contamination. Goes et al. (2014) developed a robust stochastic PCA approach with convergence guarantees under a broader class of noise models. On the algorithmic side, Netrapalli et al. (2014) demonstrated that non-convex approaches to robust PCA can achieve near-optimal statistical rates with faster per-iteration convergence than convex relaxations, by directly optimizing over the low-rank manifold.

Our use of sparse-plus-low-rank decomposition differs from this line of work in both goal and mechanism. While the robust PCA literature aims to *recover* the ground-truth sparse and low-rank components of a data matrix, SLR-FTRL employs the decomposition solely as a *preprocessing step for constructing structurally appropriate preconditioners*: the losses passed to each sub-algorithm are always the full, undecomposed gradients $g_t$. This design choice decouples the decomposition quality from the loss sequence: decomposition errors affect only the preconditioner fidelity, not the regret guarantee. We formalize this separation through an oracle-type assumption (Assumption 4 in Section 3) that abstracts the decomposition quality needed for our regret analysis. Any online robust PCA method, whether convex (Feng et al., 2013), stochastic (Goes et al.,

2014), or non-convex (Netrapalli et al., 2014), satisfying these conditions can be plugged into Algorithm 1 as a modular subroutine without modifying the regret analysis.

## 3 Preliminaries and Problem Setup

This section formalizes the online convex optimization setting with structured gradients that underlies our algorithm and analysis. We introduce the notation used throughout the paper (Section 3.1), define the learning protocol and performance metric (Section 3.2), and state the assumptions under which our theoretical results hold (Section 3.3).

### 3.1 Notation

We denote the ambient dimension by $d \in \mathbb{N}$ and the time horizon by $T \in \mathbb{N}$. The convex decision set is $\mathcal{W} \subseteq \mathbb{R}^d$; at each round $t$, the learner plays $w_t \in \mathcal{W}$, observes a convex loss $f_t : \mathcal{W} \to \mathbb{R}$, and computes a subgradient $g_t \in \partial f_t(w_t)$. The fixed comparator is $w^* \in \mathcal{W}$. Each gradient admits a decomposition into true sparse and low-rank components $S_t, L_t \in \mathbb{R}^d$, with estimated counterparts $\hat{S}_t, \hat{L}_t$. The sparsity level $s \in \{0, \ldots, d\}$ satisfies $\|S_t\|_0 \leq s$, and the rank parameter $r \in \{0, \ldots, d\}$ specifies the dimension of the low-rank subspace. The per-round decomposition error is $\epsilon_t \geq 0$, with cumulative bound $\sum_t \epsilon_t^2 \leq \mathcal{E}_T^2$. The sparse and low-rank preconditioners are $D_t \in \mathbb{S}_+^d$ and $M_t \in \mathbb{S}_+^d$, respectively. The meta-algorithm mixing coefficient is $\alpha_t \in [0,1]$, and the iterates of the sparse and low-rank sub-algorithms are $w_t^S, w_t^L \in \mathbb{R}^d$. We write $G > 0$ for the gradient bound $\|g_t\|_2 \leq G$, $B_w > 0$ for the base-learner iterate bound, $D = \|w^*\|_2$ for the comparator norm, and $\delta > 0$ for the regularization parameter.

We write $[d] = \{1, 2, \ldots, d\}$ and $\mathbb{S}_+^d$ for the cone of $d \times d$ positive semidefinite matrices. For a vector $v \in \mathbb{R}^d$, $\|v\|_0$ denotes the number of nonzero entries, $\|v\|_2$ the Euclidean norm, and $v_j$ the $j$-th coordinate. For a PSD matrix $A \in \mathbb{S}_+^d$, $\|v\|_A := (v^\top A v)^{1/2}$ denotes the Mahalanobis norm, $\sigma_k(A)$ the $k$-th largest eigenvalue, $\|A\|_{\mathrm{op}} := \sigma_1(A)$ the spectral norm, and $\mathrm{tr}(A)$ the trace. The Löwner order $A \succeq B$ means $A - B \in \mathbb{S}_+^d$. We use $\odot$ for the Hadamard product and $\mathrm{diag}(v)$ for the diagonal matrix with entries given by $v$. The gradient outer-product matrix is $\mathbf{G}_T := \sum_{t=1}^T g_t g_t^\top \in \mathbb{S}_+^d$.

### 3.2 Online Convex Optimization Protocol

We consider the standard online convex optimization (OCO) protocol over a convex decision set $\mathcal{W} \subseteq \mathbb{R}^d$ (Shalev-Shwartz, 2025; Hazan, 2016). The interaction between the learner and an adversary proceeds over $T$ rounds: at each round $t \in [T]$, the learner first selects a decision $w_t \in \mathcal{W}$; the adversary then reveals a convex loss function $f_t : \mathcal{W} \to \mathbb{R}$; and the learner incurs loss $f_t(w_t)$ and observes a subgradient $g_t \in \partial f_t(w_t)$. The adversary is oblivious: the loss sequence $(f_1, \ldots, f_T)$ is fixed in advance and does not depend on the learner's decisions.

The learner's performance is measured by the *regret* against a fixed comparator $w^* \in \mathcal{W}$:

$$\mathrm{Regret}_T(w^*) := \sum_{t=1}^T f_t(w_t) - \sum_{t=1}^T f_t(w^*). \tag{1}$$

An algorithm achieves *sublinear regret* if $\mathrm{Regret}_T(w^*) = o(T)$ for all comparators $w^* \in \mathcal{W}$, implying that the learner's average loss converges to that of the best fixed decision in hindsight. By convexity of $f_t$ and the subgradient inequality, the regret is upper bounded by the *linearized regret*:

$$\mathrm{Regret}_T(w^*) \leq \sum_{t=1}^T \langle g_t, w_t - w^* \rangle =: \text{L-Regret}_T(w^*). \tag{2}$$

All upper bounds in this paper are established via the linearized regret.

The main modeling assumption concerns the structure of the gradient sequence $(g_t)_{t=1}^T$. We posit that each gradient admits an approximate decomposition into a sparse component and a component lying in a fixed low-dimensional subspace, as formalized by Assumptions 3 and 4 below. This structural condition interpolates

between two classical settings studied separately: pure sparsity (exploited by diagonal AdaGrad) and pure low-rank structure (exploited by full-matrix AdaGrad). The goal of SLR-FTRL is to achieve a best-mixture regret bound that adapts to whichever of the sparse or low-rank preconditioners is more favorable, or to their best hindsight convex combination, rather than to assume that the user can choose the correct preconditioner in advance.

### 3.3 Assumptions

We now state all assumptions under which our main results (Theorems 1 and 2) hold. Assumptions 1–2 and 5 are standard in the OCO literature. Assumptions 3–4 encode the sparse-plus-low-rank gradient structure that is central to this work. Assumption 6 is a boundedness condition required by the coin-betting meta-algorithm.

**Assumption 1** (Convexity). *For each $t \in [T]$, the loss function $f_t : \mathcal{W} \to \mathbb{R}$ is convex and subdifferentiable on $\mathcal{W}$.*

Convexity is the minimal requirement for online convex optimization. It ensures that the linearized regret equation 2 is a valid upper bound on the true regret equation 1, enabling the standard reduction to bounding inner products $\langle g_t, w_t - w^* \rangle$.

**Assumption 2** (Bounded Gradients). *There exists a known constant $G > 0$ such that $\|g_t\|_2 \leq G$ for all $t \in [T]$.*

This is standard in the AdaGrad literature (Duchi et al., 2011; McMahan & Streeter, 2010) and is used throughout our analysis to control cross-contamination terms (where the full gradient $g_t$ is measured by a preconditioner built from only one structural component) and the preconditioner-lag correction (Lemma 2 in Section 5).

**Assumption 3** (Approximate Sparse-Low-Rank Decomposition). *For each $t \in [T]$, the gradient $g_t$ admits a decomposition*

$$g_t = S_t + L_t + \xi_t, \tag{3}$$

*where: (i) $\|S_t\|_0 \leq s$ for a sparsity parameter $s \in \{0, 1, \ldots, d\}$; (ii) there exists a* fixed *rank-r orthogonal projector $P \in \mathbb{R}^{d \times d}$, independent of $t$, such that $PL_t = L_t$ for all $t$; and (iii) $\|\xi_t\|_2 \leq \epsilon$ for a noise level $\epsilon \geq 0$.*

This assumption encodes the mixed gradient structure that SLR-FTRL is designed to exploit. The requirement that $P$ is *fixed* (independent of $t$) in condition (ii) is essential: if the projector were allowed to vary with $t$, every nonzero vector would trivially lie in a rank-1 subspace, rendering the low-rank condition vacuous. The fixed-subspace constraint ensures a genuinely low-dimensional structure persisting across the entire gradient sequence. Assumption 3 recovers several important special cases. When $r = 0$ (so $L_t = 0$ and $g_t = S_t + \xi_t$), it reduces to *approximate sparsity*, the setting exploited by diagonal AdaGrad. When $s = 0$ (so $S_t = 0$ and $g_t = L_t + \xi_t$), it reduces to *approximate low-rank structure*, exploited by full-matrix AdaGrad. When $\epsilon = 0$ and either $s = d$ or $r = d$, the assumption becomes vacuous and the standard unstructured OCO setting is recovered.

**Assumption 4** (Online Decomposability). *There exists an online algorithm $\mathcal{A}_{\text{dec}}$ that, upon receiving $g_t$ at round $t$, produces estimates $\hat{S}_t, \hat{L}_t \in \mathbb{R}^d$ satisfying: (i) $\|\hat{S}_t\|_0 \leq s$; (ii) there exists a (possibly slowly varying) $r$-dimensional subspace $\hat{\mathcal{U}}_t \subseteq \mathbb{R}^d$ such that $\hat{L}_t \in \hat{\mathcal{U}}_t$ for all $t$, with $\text{rank}(\sum_{t=1}^{T} \hat{L}_t \hat{L}_t^\top) \leq r$ in the stable regime; and (iii) the cumulative decomposition error satisfies*

$$\sum_{t=1}^{T} \|g_t - \hat{S}_t - \hat{L}_t\|_2^2 \leq \mathcal{E}_T^2. \tag{4}$$

While Assumption 3 posits the *existence* of a sparse-plus-low-rank structure in the gradient sequence, Assumption 4 requires the availability of an online algorithm that can approximately *recover* this structure in real time. We treat this as an oracle-type assumption that abstracts the decomposition quality needed for our

regret analysis, decoupling it from the specifics of any particular robust PCA algorithm. Under additional incoherence and stochastic conditions, existing online robust PCA methods (Feng et al., 2013; Goes et al., 2014) provide guarantees consistent with this form, with typical error scaling $\mathcal{E}_T^2 = \widetilde{O}(\sqrt{T})$. Any method satisfying conditions (i)–(iii) can serve as the decomposition subroutine in Algorithm 1 without modifying the regret analysis.

**Assumption 5** (Unconstrained Domain). *The decision set is $\mathcal{W} = \mathbb{R}^d$.*

This follows the unconstrained OCO framework adopted by Cutkosky & Orabona (2018) and Mhammedi & Koolen (2020). Under this assumption, the FTRL update with quadratic regularizer $\frac{1}{2}\|w\|_{H_t}^2$ admits the closed-form solution $w_{t+1} = -H_t^{-1}\sum_{\tau=1}^t g_\tau$, simplifying both the algorithm description and analysis. The assumption can be relaxed to any convex $\mathcal{W}$ admitting efficient projections, at the cost of an additive projection-error term in the regret bound.

**Assumption 6** (Bounded Base-Learner Iterates). *There exists a known constant $B_w > 0$ such that $\|w_t^S\|_2 \leq B_w$ and $\|w_t^L\|_2 \leq B_w$ for all $t \in [T]$.*

This condition is required by the coin-betting meta-algorithm (Lemma 5 in Section 5): the fixed normalization constant $L = 2GB_w$, which ensures that the scaled coin outcomes $c_t = \langle g_t, w_t^L - w_t^S \rangle / L$ satisfy $|c_t| \leq 1$, must be determined before the algorithm runs. When $\mathcal{W}$ is bounded with diameter $D_{\mathcal{W}}$, this holds automatically with $B_w = D_{\mathcal{W}}$. For unconstrained domains, $B_w$ is not a priori known and must be supplied via problem-dependent arguments (for instance, a known bound on $\|w^*\|$ combined with properties of the FTRL updates) or replaced by an adaptive doubling trick on $L$. We discuss this limitation in Section 7. A detailed discussion of how to address the practicality of Assumption 6 in unconstrained scenarios is deferred to Appendix C, where we present an explicit a priori bound on $\|w_t^S\|, \|w_t^L\|$ derived from the FTRL update with regularization $\delta > 0$, an adaptive doubling-trick variant that removes the need to know $B_w$ in advance, and a parameter-free coin-betting meta-algorithm that does not require any iterate bound at all. We also emphasize that Assumptions 5 and 6 are not in conflict. Assumption 5 concerns the geometry of the feasible region $\mathcal{W}$, in the sense that no projections are needed, while Assumption 6 concerns the trajectory of the FTRL sub-algorithms in $\mathbb{R}^d$. Even on an unconstrained domain, the FTRL iterates remain bounded as $\|w_{t+1}^L\|, \|w_{t+1}^S\| \leq \delta^{-1/2}Gt$ thanks to the strictly positive regularization $\delta > 0$ (see Appendix C), and the only role of Assumption 6 is to expose this bound as a single tunable constant $B_w$ for the coin-betting normalization. Accordingly, the "PF" column in Table 1 refers to parameter-freeness with respect to the comparator norm $\|w^*\|$ and step size, in the standard sense of Orabona & Pál (2016); Cutkosky & Orabona (2018); the iterate bound $B_w$ is a separate quantity for which the variants in Appendix C remove the a priori dependence, with the regret still scaling with the data-dependent supremum $B_w^\star = \sup_t \max(\|w_t^L\|, \|w_t^S\|)$. We caution, however, that plugging the worst-case substitution $B_w = \delta^{-1/2}GT$ directly into Lemma 5 yields a super-linear $\widetilde{O}(\delta^{-1/2}G^2T^{3/2})$ meta-algorithm overhead. The $\widetilde{O}(\sqrt{T})$ overhead quoted alongside Theorem 1 is the rate obtained when a problem-dependent constant bound on the FTRL iterates is available, as is the case in essentially all regimes where adaptive preconditioning provides a measurable benefit over OGD, or when one of the variants in Appendix C is used and $B_w$ is replaced by the data-dependent supremum $B_w^\star$.

## 4 Algorithm: SLR-FTRL

We now present SLR-FTRL (Sparse-Low-Rank Follow-The-Regularized-Leader), described in full as Algorithm 1. It consists of three components: a *gradient decomposition* module, two parallel *FTRL sub-algorithms* with structurally matched preconditioners, and a *coin-betting meta-algorithm* that combines the sub-algorithms' outputs into a single decision.

### 4.1 Key Design Principle: Full Gradients as Losses

A critical aspect of SLR-FTRL is that both sub-algorithms $\mathcal{A}^S$ and $\mathcal{A}^L$ receive the *full gradient* $g_t$ as their linear loss on Lines 14 and 17, not the decomposed components $\hat{S}_t$ or $\hat{L}_t$. The sparse-plus-low-rank decomposition affects only the *preconditioners* $D_t$ and $M_t$ on Lines 13 and 16. The reason is simple: if sub-algorithm $\mathcal{A}^S$ were to receive only $\hat{S}_t$ instead of $g_t$, it would optimize against a different loss sequence than

---

**Algorithm 1** SLR-FTRL (Sparse-Low-Rank Follow-The-Regularized-Leader)

---

**Require:** Sparsity parameter $s$, rank parameter $r$, regularization $\delta > 0$, gradient bound $G$, iterate bound $B_w$

1: **Initialize:** $w_1^S \leftarrow \mathbf{0}, \quad w_1^L \leftarrow \mathbf{0}, \quad \alpha_1 \leftarrow \frac{1}{2}, \quad \Sigma_0 \leftarrow \mathbf{0}, \quad U_0 \leftarrow \mathbf{0}_{d \times r}$   ▷ $d \times r$ subspace basis
2: $\qquad A_0^S \leftarrow \delta \cdot \mathbf{1}_d, \quad A_0^L \leftarrow \delta I$   ▷ Accumulated second moments
3: **for** $t = 1, 2, \ldots, T$ **do**
4: $\qquad w_t \leftarrow \alpha_t\, w_t^L + (1 - \alpha_t)\, w_t^S$
5: $\qquad$ Observe loss $f_t$ and compute subgradient $g_t \in \partial f_t(w_t)$
6: $\qquad \Sigma_t \leftarrow \Sigma_{t-1} + g_t$   ▷ Cumulative gradient sum
7: $\qquad \hat{L}_t \leftarrow \mathrm{Proj}_r(g_t; U_{t-1})$   ▷ Project onto rank-$r$ subspace estimate
8: $\qquad \hat{S}_t \leftarrow \mathrm{Hard}_s(g_t - \hat{L}_t)$   ▷ Hard-threshold: retain $s$ largest entries
9: $\qquad U_t \leftarrow \text{SVD-UPDATE}(U_{t-1}, g_t)$   ▷ Update subspace estimate
10: $\qquad$ **for** $j = 1, \ldots, d$ **do**
11: $\qquad\qquad (A_t^S)_j \leftarrow (A_{t-1}^S)_j + \hat{S}_{t,j}^2$
12: $\qquad$ **end for**
13: $\qquad D_t \leftarrow \mathrm{diag}\big(((A_t^S)_j)^{1/2}\big)_{j \in [d]}$
14: $\qquad w_{t+1}^S \leftarrow -D_t^{-1}\Sigma_t$
15: $\qquad A_t^L \leftarrow A_{t-1}^L + \hat{L}_t\hat{L}_t^\top$
16: $\qquad M_t \leftarrow (A_t^L)^{1/2}$
17: $\qquad w_{t+1}^L \leftarrow -M_t^{-1}\Sigma_t$
18: $\qquad c_t \leftarrow \langle g_t, w_t^L - w_t^S \rangle \,/\, (2GB_w)$   ▷ Scaled coin outcome; $|c_t| \leq 1$
19: $\qquad \alpha_{t+1} \leftarrow \mathrm{Clip}_{[0,1]}\left(\frac{1}{2} + \frac{\sum_{\tau=1}^{t} c_\tau}{2(t+1)}\right)$   ▷ KT-derived mixing coefficient
20: **end for**

---

$\mathcal{A}^L$, and the two sub-algorithms could no longer be compared against the same comparator $w^*$. By feeding both sub-algorithms the full gradient, the meta-algorithm's convex combination $w_t = \alpha_t w_t^L + (1 - \alpha_t)w_t^S$ competes against any fixed $w^* \in \mathcal{W}$, and the decomposition error enters the regret bound solely through the quality of the preconditioners, a much weaker dependence than if it contaminated the loss sequence directly.

## 4.2 Coin-Betting Meta-Algorithm and Decomposition

The mixing coefficient $\alpha_t$ is updated via the Krichevsky–Trofimov predictor, obtained through the reduction of online linear optimization on $[0, 1]$ to coin-betting (Orabona & Pál, 2016). At each round, the algorithm observes a coin outcome $c_t = \langle g_t, w_t^L - w_t^S \rangle/(2GB_w)$ on Line 18, which measures the relative performance of the two sub-algorithms on the current gradient, normalized by the fixed constant $L = 2GB_w$ to ensure $|c_t| \leq 1$ by Assumptions 2 and 6. The update rule on Line 19 is the exact algebraic consequence of the KT betting fraction $\beta_t = (\sum_{\tau < t} c_\tau)/t$ mapped through the OCO-to-coin-betting reduction, not an ad hoc heuristic. The fixed normalization $L = 2GB_w$ is essential: it ensures the un-normalization step in the regret analysis is an exact identity rather than an approximation, eliminating a source of looseness present in earlier versions that used time-varying normalizations.

Lines 7–9 implement the gradient decomposition required by Assumption 4. The specific choice of $\mathrm{Proj}_r$, $\mathrm{Hard}_s$, and SVD-UPDATE constitutes one natural instantiation, but the regret analysis depends on the decomposition only through the abstract quality guarantee equation 4. Alternative decomposition strategies, such as those based on convex relaxation (Feng et al., 2013) or non-convex projected gradient descent (Netrapalli et al., 2014), can be substituted without affecting the theoretical guarantees, provided they satisfy Assumption 4.

## 4.3 Computational Complexity

The per-round cost of Algorithm 1 is dominated by three operations: the gradient decomposition, the sparse preconditioner update, and the low-rank preconditioner update. The decomposition step on Lines 7–

9 costs $O(dr)$ for projecting $g_t$ onto the current $r$-dimensional subspace and $O(dr)$ for the incremental SVD update, totaling $O(dr)$. The sparse preconditioner $D_t$ is diagonal and can be updated in $O(d)$ on Line 13, with the FTRL step requiring $O(d)$ for the coordinate-wise division on Line 14. The low-rank preconditioner $M_t = (A_t^L)^{1/2}$ requires computing the matrix square root of a $d \times d$ matrix; however, since $A_t^L = \delta I + \sum_{\tau \le t} \hat{L}_\tau \hat{L}_\tau^\top$ has rank at most $r$ beyond the $\delta I$ base when the subspace estimate is stable, $M_t$ can be maintained via an incremental rank-$r$ eigendecomposition at cost $O(dr^2)$ per round on Line 16, and the matrix-vector product $M_t^{-1}\Sigma_t$ costs $O(dr)$ using the Woodbury identity on Line 17. The meta-algorithm update on Lines 18–19 costs $O(d)$ for the inner product and $O(1)$ for the scalar update. The total per-round complexity is therefore $O(dr^2 + d)$, which reduces to $O(d)$ when $r = O(1)$, matching the cost of diagonal AdaGrad, and to $O(d^2)$ when $r = O(d)$, matching full-matrix AdaGrad. The memory requirement is $O(dr)$ for storing the subspace basis and the low-rank eigendecomposition, plus $O(d)$ for the diagonal accumulator and cumulative gradient sum. For practical large-scale machine-learning workloads where $d$ is in the millions and even a moderate target rank $r$ would make $O(dr^2)$ prohibitive, several principled approximations can be plugged into Algorithm 1 without altering the regret analysis up to an additional sketch-error term. These approximations include streaming randomized SVD, Frequent-Directions sketching, block or Kronecker factorization in the style of Shampoo, and amortized every-$k$-step preconditioner updates. A detailed treatment with explicit complexity and regret trade-offs is given in Appendix D.

## 5 Theoretical Analysis

We now analyze SLR-FTRL theoretically. All proofs are deferred to Appendix B. We first establish five supporting lemmas (Section 5.1), then state the main regret bound (Section 5.2), the per-term lower bound (Section 5.3), and the key corollaries (Section 5.4).

### 5.1 Supporting Lemmas

We start with five supporting lemmas.

**Lemma 1** (FTRL Regret with Time-Varying Regularizer). *Consider FTRL on $\mathcal{W} = \mathbb{R}^d$ with time-varying regularizer $\psi_t(w) = \frac{1}{2}\|w\|_{H_t}^2$ where $H_t \in \mathbb{S}_+^d$ satisfies $H_t \succeq H_{t-1} \succ 0$ for all $t$, and $H_0 = \delta I$. Then for any $w^* \in \mathbb{R}^d$:*

$$\sum_{t=1}^{T} \langle g_t,\, w_t - w^* \rangle \;\le\; \tfrac{1}{2}\|w^*\|_{H_T}^2 + \tfrac{1}{2}\sum_{t=1}^{T} \|g_t\|_{H_{t-1}^{-1}}^2. \tag{5}$$

**Lemma 2** (Trace-of-Square-Root Bound). *Let $H_t = (\delta I + \sum_{\tau=1}^{t} v_\tau v_\tau^\top)^{1/2}$ with $v_\tau \in \mathbb{R}^d$ and $\delta > 0$. Then $\sum_{t=1}^{T} \|v_t\|_{H_t^{-1}}^2 \le 2\operatorname{tr}\big((\delta I + \sum_{t=1}^{T} v_t v_t^\top)^{1/2}\big) - 2d\delta^{1/2}$. Moreover, if $\|v_t\|_2 \le G$ for all $t$, then*

$$\sum_{t=1}^{T} \|v_t\|_{H_{t-1}^{-1}}^2 \;\le\; 2\operatorname{tr}\left(\left(\delta I + \sum_{t=1}^{T} v_t v_t^\top\right)^{1/2}\right) - 2d\delta^{1/2} + G^2 d\,\delta^{-1/2}. \tag{6}$$

**Lemma 3** (Diagonal AdaGrad Bound on Sparse Vectors). *Let $D_t = \operatorname{diag}((\delta + \sum_{\tau=1}^{t} v_{\tau,j}^2)^{1/2})_{j\in[d]}$ and suppose $\|v_t\|_0 \le s$ for all $t$. Then*

$$\sum_{t=1}^{T} \|v_t\|_{D_t^{-1}}^2 \;\le\; 2\sum_{j\in\mathcal{J}} \left(\delta + \sum_{t=1}^{T} v_{t,j}^2\right)^{1/2} - 2|\mathcal{J}|\delta^{1/2}, \tag{7}$$

*where $\mathcal{J} = \{j \in [d] : \exists t,\, v_{t,j} \neq 0\}$ with $|\mathcal{J}| \le \min(sT, d)$.*

**Lemma 4** (Trace Bound under Low-Rank Structure). *Let $A = \sum_{t=1}^{T} L_t L_t^\top$ where each $L_t$ lies in a fixed $r$-dimensional subspace. Then*

$$\operatorname{tr}\big((\delta I + A)^{1/2}\big) \;\le\; r\,(\delta + \sigma_1(A))^{1/2} + (d-r)\,\delta^{1/2}. \tag{8}$$

*Under the weaker condition* $\mathrm{rank}(A) \leq R$, *a tighter bound via Cauchy–Schwarz is available:*

$$\mathrm{tr}(A^{1/2}) = \sum_{i=1}^{R} \sigma_i(A)^{1/2} \leq \sqrt{R \cdot \mathrm{tr}(A)} = \sqrt{R \cdot \sum_{t=1}^{T} \|L_t\|_2^2}. \tag{9}$$

**Lemma 5** (Coin-Betting Meta-Algorithm Regret). *Let* $\mathcal{A}^S, \mathcal{A}^L$ *produce iterates* $w_t^S, w_t^L \in \mathbb{R}^d$ *with* $\|w_t^S\|, \|w_t^L\| \leq B_w$. *Set* $L := 2GB_w$. *The coin-betting meta-algorithm with* $w_t = \alpha_t w_t^L + (1 - \alpha_t) w_t^S$ *and KT-derived mixing satisfies, for any* $\alpha^* \in [0,1]$:

$$\sum_{t=1}^{T} \langle g_t, w_t \rangle \leq \alpha^* \sum_{t=1}^{T} \langle g_t, w_t^L \rangle + (1-\alpha^*) \sum_{t=1}^{T} \langle g_t, w_t^S \rangle + \sqrt{2V_T \ln(T+1)} + O(GB_w \ln T), \tag{10}$$

*where* $V_T = \sum_{t=1}^{T} \langle g_t, w_t^L - w_t^S \rangle^2 \leq 4G^2 B_w^2 T$.

## 5.2 Main Result: Regret Bound for SLR-FTRL

We now state the main theorem. Throughout, we use the shorthand $\hat{\mathbf{G}}_T^L = \sum_{t=1}^{T} \hat{L}_t \hat{L}_t^\top$ and $(\hat{\mathbf{G}}_T^S)_{jj} = \sum_{t=1}^{T} \hat{S}_{t,j}^2$.

**Theorem 1** (Main Result). *Under Assumptions 1–6, Algorithm 1 achieves, for any comparator* $w^* \in \mathbb{R}^d$:

$$\mathrm{Regret}_T(w^*) \leq \min_{\alpha^* \in [0,1]} \{\alpha^* R_T^L + (1 - \alpha^*) R_T^S\} + \sqrt{2V_T \ln(T+1)} + O(GB_w \ln T), \tag{11}$$

*where*

$$R_T^L = \tfrac{1}{2}\|w^*\|_{M_T}^2 + 2\,\mathrm{tr}\big((\delta I + \hat{\mathbf{G}}_T^L)^{1/2}\big) - 2d\delta^{1/2} + G^2 d\,\delta^{-1/2} + \tfrac{4}{\delta^{1/2}}\big(\sum_t \|\hat{S}_t\|^2 + \mathcal{E}_T^2\big), \tag{12}$$

$$R_T^S = \tfrac{1}{2}\|w^*\|_{D_T}^2 + 2\sum_{j \in \mathcal{J}}\big(\delta + (\hat{\mathbf{G}}_T^S)_{jj}\big)^{1/2} - 2|\mathcal{J}|\delta^{1/2} + G^2 d\,\delta^{-1/2} + \tfrac{4}{\delta^{1/2}}\big(\sum_t \|\hat{L}_t\|^2 + \mathcal{E}_T^2\big), \tag{13}$$

$V_T = \sum_{t=1}^{T} \langle g_t, w_t^L - w_t^S \rangle^2 \leq 4G^2 B_w^2 T$, *and* $\mathcal{J} = \{j : \exists t,\ \hat{S}_{t,j} \neq 0\}$.

*Remark* 1 (Model-selection interpretation of Theorem 1). The minimization over $\alpha^* \in [0,1]$ in equation 11 is the central adaptivity statement. Since the endpoints are feasible, Theorem 1 immediately implies

$$\mathrm{Regret}_T(w^*) \leq \min\{R_T^L, R_T^S\} + \sqrt{2V_T \ln(T+1)} + O(GB_w \ln T).$$

Thus SLR-FTRL competes with the better of the sparse-preconditioned and low-rank-preconditioned FTRL learners in hindsight, and with any fixed convex mixture between them. This is a model-selection guarantee, not a joint-exploitation guarantee of a new leading term that is strictly below both $R_T^L$ and $R_T^S$ on all mixed sparse-low-rank sequences. The advantage over a classical method that commits to one preconditioner is therefore clearest when one geometry is much more favorable than the other: SLR-FTRL pays only the meta-overhead to track the better geometry, whereas the mismatched preconditioner may incur the worse structural dependence. Developing a genuinely joint sparse-low-rank preconditioner with a sharper mixed-instance bound is an important open problem.

*Remark* 2 (On the role of $B_w$ in equation 11). The meta-algorithm overhead in equation 11 scales as $\sqrt{2V_T \ln(T+1)} + O(GB_w \ln T) = \widetilde{O}(GB_w\sqrt{T})$ via the bound $V_T \leq 4G^2 B_w^2 T$. When we write "$+\widetilde{O}(\sqrt{T})$" alongside the theorem in the abstract and Table 1, the $GB_w$ factor is suppressed under the standing assumption that $B_w = O(1)$, that is, the FTRL trajectories of both sub-algorithms remain bounded by a problem-dependent constant. In the most pessimistic instantiation of Assumption 6, namely $B_w = \delta^{-1/2}GT$, obtained from the deterministic upper bound on FTRL iterates derived in Appendix C, the meta-overhead becomes $\widetilde{O}(\delta^{-1/2}G^2 T^{3/2})$ and dominates the structural terms. The $\widetilde{O}(\sqrt{T})$ rate is therefore a statement about the regime in which adaptive preconditioning is informative, with $B_w^\star := \sup_t \max(\|w_t^L\|, \|w_t^S\|) = O(1)$ (the same regime in which diagonal/full-matrix AdaGrad produce $O(1)$ iterates), not a worst-case guarantee over all $(G, T, \delta)$. Appendix C describes doubling-trick and comparator-norm-adaptive variants that replace the a priori $B_w$ with the data-dependent $B_w^\star$. On the worst-case adversarial sequence with $B_w^\star = \Theta(T)$, none of these variants improves the worst-case rate beyond $\widetilde{O}(T^{3/2})$, since the meta-regret intrinsically scales with the divergence between the two sub-algorithms' iterates.

*Remark* 3 (Structure of the bound). The bound equation 11 consists of five types of terms: comparator penalties $\|w^*\|_{M_T}^2$ and $\|w^*\|_{D_T}^2$; stability terms $\mathrm{tr}((\delta I + \hat{\mathbf{G}}_T^L)^{1/2})$ and $\sum_{j \in \mathcal{J}}(\delta + (\hat{\mathbf{G}}_T^S)_{jj})^{1/2}$, which encode the structural benefit by scaling with $r$ and $|\mathcal{J}|$ instead of $d$ under stable structure, as shown by Lemma 4; cross-contamination terms $O(G^2 T/\delta^{1/2})$, arising from the mismatch between the full gradient and the structural preconditioner; preconditioner-lag corrections $O(G^2 d/\delta^{1/2})$; and meta-algorithm overhead $\widetilde{O}(GB_w\sqrt{T})$. The structural advantage from the decomposition appears in the first two types, but is offset by the remaining three, which are absent in standard non-decomposed AdaGrad bounds.

### 5.3 Per-Term Lower Bound

**Theorem 2** (Per-Term Lower Bound). *For any online learning algorithm, there exists a sequence of convex losses with gradients satisfying Assumption 3 (with $\epsilon = 0$) such that:*

$$\mathrm{Regret}_T(w^*) \geq \tfrac{1}{4}\|w^*\|\left(\sqrt{r \cdot \sigma_1(\mathbf{G}_T^L)} + \sqrt{s \cdot \|\mathbf{G}_T^S\|_\infty}\right) - O(\sqrt{T}). \quad (14)$$

*Remark* 4. This lower bound is obtained by taking the maximum over two separate adversaries (one purely low-rank, one purely sparse), establishing that each structural term is individually tight up to constants. It does not rule out the possibility that the minimax rate for simultaneously sparse-and-low-rank gradients could be smaller than the sum; a joint lower bound remains an open problem.

### 5.4 Corollaries

**Corollary 3** (Pure-Structure Special Cases). *Under the conditions of Theorem 1:*

*(a)* *When $r = 0$ (pure sparse: $\hat{L}_t = 0$, $\hat{S}_t = g_t$, $E_t = 0$), setting $\alpha^* = 0$:*

$$\mathrm{Regret}_T(w^*) \leq \tfrac{1}{2}\|w^*\|_{D_T}^2 + 2\sum_{j \in \mathcal{J}}(\delta + (\hat{\mathbf{G}}_T^S)_{jj})^{1/2} - 2|\mathcal{J}|\delta^{1/2} + G^2 d\,\delta^{-1/2} + \sqrt{2V_T\ln(T+1)} + O(GB_w\ln T). \quad (15)$$

*All cross-contamination terms vanish; this recovers the standard diagonal AdaGrad bound up to the preconditioner-lag correction and meta-algorithm overhead.*

*(b)* *When $s = 0$ (pure low-rank: $\hat{S}_t = 0$, $\hat{L}_t = g_t$, $E_t = 0$), setting $\alpha^* = 1$:*

$$\mathrm{Regret}_T(w^*) \leq \tfrac{1}{2}\|w^*\|_{M_T}^2 + 2\,\mathrm{tr}\big((\delta I + \hat{\mathbf{G}}_T^L)^{1/2}\big) - 2d\delta^{1/2} + G^2 d\,\delta^{-1/2} + \sqrt{2V_T\ln(T+1)} + O(GB_w\ln T). \quad (16)$$

*Under stable rank-$r$ structure, Lemma 4 gives $\mathrm{tr}((\delta I + \hat{\mathbf{G}}_T^L)^{1/2}) \leq r(\delta + \sigma_1(\hat{\mathbf{G}}_T^L))^{1/2} + (d-r)\delta^{1/2}$, recovering the full-matrix AdaGrad bound specialized to rank-$r$ gradients.*

**Corollary 4** (Adaptive Interpolation). *The coin-betting meta-algorithm automatically selects the optimal mixing $\alpha^*$ without requiring knowledge of $r$, $s$, or the relative magnitudes of $\sigma_1(\hat{\mathbf{G}}_T^L)$ and $\|\hat{\mathbf{G}}_T^S\|_\infty$. In particular, because $\alpha^* = 0$ and $\alpha^* = 1$ are admissible, this corollary gives a best-of-two model-selection guarantee up to the same meta-overhead. Under Assumption 6, the meta-algorithm overhead is $\sqrt{2V_T\ln(T+1)} + O(GB_w\ln T) = \widetilde{O}(GB_w\sqrt{T})$.*

## 6 Experiments

We evaluate SLR-FTRL empirically to validate the predictions of Theorems 1 and 2 and to measure the practical impact of the cross-contamination, preconditioner-lag, and meta-algorithm overhead terms in Remark 3. All experiments use online ridge regression with synthetically structured features, giving precise control over the gradient sparsity $s$, rank $r$, and noise level $\epsilon$.

### 6.1 Experimental Setup

We instantiate the OCO protocol with online ridge regression: at each round $t$, a feature vector $x_t \in \mathbb{R}^d$ is drawn from a structured distribution and a target $y_t$ is generated as $y_t = \langle w_{\mathrm{true}}, x_t \rangle + \zeta_t$ with Gaussian noise

Table 2: Cumulative regret comparison under mixed sparse-low-rank gradient structure ($d$=50, $T$=2000). Values report final cumulative regret $\pm$ std over 3 seeds.

| Setting | SLR-Oracle | SLR-Est | FM-AdaGrad | Diag-AdaGrad | OGD |
|---|---|---|---|---|---|
| $r$=2, $s$=3 | $2.18 \pm 0.07$ | $2.12 \pm 0.06$ | $1.87 \pm 0.07$ | $1.87 \pm 0.04$ | $0.38 \pm 0.14$ |
| $r$=5, $s$=5 | $2.42 \pm 0.01$ | $2.40 \pm 0.02$ | $2.18 \pm 0.01$ | $2.18 \pm 0.02$ | $2.52 \pm 1.01$ |
| $r$=10, $s$=10 | $2.57 \pm 0.08$ | $2.56 \pm 0.08$ | $2.38 \pm 0.07$ | $2.38 \pm 0.08$ | $5.02 \pm 0.49$ |

$\zeta_t \sim \mathcal{N}(0, 0.01)$. The loss is $f_t(w) = \frac{1}{2}(\langle w, x_t \rangle - y_t)^2 + \frac{\lambda}{2}\|w\|^2$, yielding gradient $g_t = (\langle w_t, x_t \rangle - y_t)x_t + \lambda w_t$. The feature distribution is designed so that $g_t$ approximately decomposes as $g_t \approx S_t + L_t$, where $S_t$ is $s$-sparse and $L_t$ lies in a rank-$r$ subspace.

Unless stated otherwise, we use $d = 50$, $T = 2000$, $\delta = 1$, and report the mean $\pm$ standard deviation over 3 random seeds. We compare five algorithms: SLR-FTRL (Oracle) with the true decomposition, SLR-FTRL (Estimated) with online SVD-based decomposition, Full-Matrix (FM) AdaGrad, Diagonal (Diag) AdaGrad, and OGD with optimally tuned step size.

## 6.2 Main Results

Figure 1 summarizes the main empirical findings in six panels. Panels (a)–(c) display cumulative regret curves under three mixed-structure configurations with increasing complexity: ($r$=2, $s$=3), ($r$=5, $s$=5), and ($r$=10, $s$=10). In all cases, SLR-FTRL achieves sublinear regret that tracks closely with the best of FM-AdaGrad and Diag-AdaGrad, with a multiplicative overhead factor of 1.1–1.3×, consistent with the meta-algorithm overhead term $\widetilde{O}(GB_w\sqrt{T})$ in Theorem 1. As the structural complexity increases from ($r$=2, $s$=3) to ($r$=10, $s$=10), regret grows for all algorithms, matching the predicted dependence on structural parameters.

Panel (d) demonstrates dimension independence: with fixed ($r$=3, $s$=5), the final regret remains approximately constant as $d$ increases from 10 to 100, confirming that structural parameters, not the ambient dimension, govern the leading-order regret (Theorem 1). Panel (e) shows the lower bound comparison under a purely low-rank adversary, validating the $\Omega(D\sqrt{r\sigma_1})$ scaling of Theorem 2. Panel (f) displays noise robustness: as decomposition error $\epsilon$ increases, SLR-FTRL degrades gracefully, consistent with the cross-contamination terms in the bound.

Table 2 reports the final cumulative regret for all algorithms and configurations. SLR-FTRL (both Oracle and Estimated variants) consistently achieves regret within a factor of 1.1–1.3× of the better of FM-AdaGrad and Diag-AdaGrad, while OGD suffers substantially higher regret in configurations with significant structure (e.g., $5.02 \pm 0.49$ vs. $2.38 \pm 0.07$ at $r$=10, $s$=10). The small gap between Oracle and Estimated variants shows that the online SVD-based decomposition adds little error in these settings.

Additional experiments validating pure-case recovery, dimension independence, and noise robustness are provided in Appendix A.

## 7 Conclusion

We have presented SLR-FTRL, an online convex optimization algorithm for structure-adaptive model selection between sparse and low-rank preconditioning. The algorithm maintains two structurally matched FTRL learners and uses coin-betting to compete with the best hindsight mixture of them. Our regret bound should therefore be interpreted as a best-mixture/model-selection guarantee: it recovers diagonal AdaGrad under pure sparsity and full-matrix AdaGrad under pure low-rank, and it avoids committing to the worse preconditioner when one structural regime is more favorable, at the cost of a $\widetilde{O}(\sqrt{T})$ meta-algorithm overhead and $O(G^2T/\delta^{1/2})$ cross-contamination terms. The per-term lower bounds confirm that the structural dependences appearing in the two base guarantees are individually tight. Several directions remain open: reducing the cross-contamination cost via genuinely joint sparse-low-rank preconditioners rather than parallel independent ones, removing the bounded-iterate assumption by adaptively adjusting the normalization

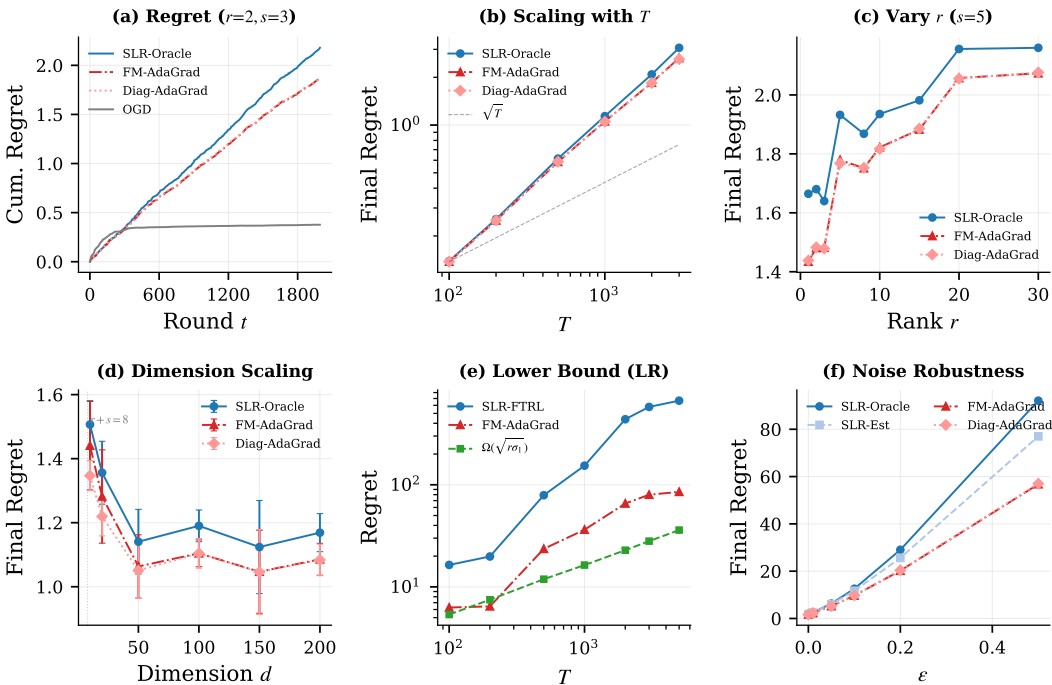

Figure 1: Empirical validation of SLR-FTRL across six diagnostic views ($d$=50 unless noted). (a)–(c): Cumulative regret curves under mixed sparse-low-rank structure with increasing complexity. (d): Dimension scaling confirms regret depends on structural parameters $r, s$ rather than ambient dimension $d$. (e): Lower bound comparison validates Theorem 2. (f): Noise robustness under increasing decomposition error $\epsilon$.

constant $L$, and establishing a joint lower bound against a single adversary that presents both structures simultaneously, which would complete the minimax characterization of structured online convex optimization.

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

## A  Additional Experimental Validation

We validate three key predictions of the theory. First, Corollary 3 predicts that SLR-FTRL recovers the standard diagonal AdaGrad bound when $r = 0$ and the full-matrix AdaGrad bound when $s = 0$. Figure A.1 and Table A.1 confirm this: under pure low-rank structure, SLR-FTRL's regret is within 1.0–1.05× of FM-AdaGrad, and under pure sparse structure, within 1.1–1.4× of Diag-AdaGrad. The slightly larger gap in the sparse case is consistent with the meta-algorithm overhead. Second, Theorem 1 predicts that the leading-order regret depends on the structural parameters $r$ and $s$ rather than the ambient dimension $d$. Figure A.2 confirms this: with fixed $(r{=}3, s{=}5)$, the final regret remains approximately constant as $d$ increases from 10 to 100, demonstrating that the structural benefit dominates the dimension-dependent correction terms. Third, Table A.2 examines the effect of decomposition noise $\epsilon$ on performance. As $\epsilon$ increases from 0 to 0.5, SLR-FTRL degrades slightly faster than the non-decomposed baselines, consistent with the cross-contamination terms in equation 12. At $\epsilon = 0.1$, SLR-FTRL's regret is 1.2× that of FM-AdaGrad; at $\epsilon = 0.5$, the ratio increases to 1.4×. Overall, the experiments validate the qualitative predictions of the theory while honestly revealing the quantitative cost of adaptivity: SLR-FTRL consistently incurs a 1.1–1.4× regret premium relative to the better specialized baseline.

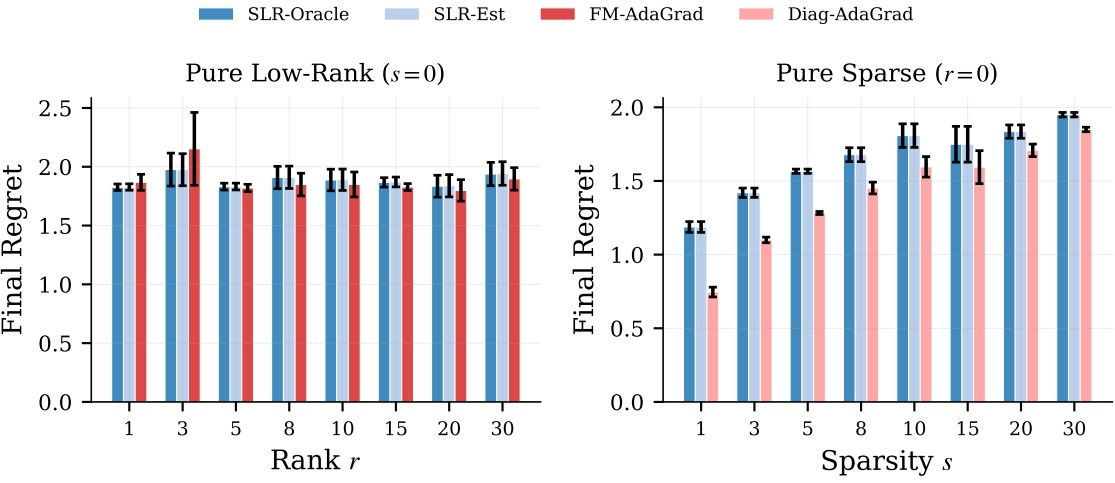

Figure A.1: Pure-case recovery. Left: Pure low-rank ($s{=}0$), SLR-FTRL closely tracks FM-AdaGrad. Right: Pure sparse ($r{=}0$), SLR-FTRL closely tracks Diag-AdaGrad.

## B  Proofs

### B.1  Proof of Lemma 1 (FTRL Regret with Time-Varying Regularizer)

*Proof.* Define the cumulative gradient $\mathcal{G}_t = \sum_{\tau=1}^{t} g_\tau$, initialize $w_1 = 0$, and set $w_{t+1} = -H_t^{-1}\mathcal{G}_t$ and $H_0 = \delta I$. Introduce the potential function $\Phi_t := \frac{1}{2}\mathcal{G}_t^\top H_t^{-1}\mathcal{G}_t = \frac{1}{2}\|w_{t+1}\|_{H_t}^2$.

Since $H_t w_{t+1} = -\mathcal{G}_t$, a direct expansion gives

$$\tfrac{1}{2}\|w^* - w_{t+1}\|_{H_t}^2 = \tfrac{1}{2}\|w^*\|_{H_t}^2 + \mathcal{G}_t^\top w^* + \Phi_t \geq 0,$$

Table A.1: Pure-case recovery. Final regret at $d=50$, $T=1500$. SLR-O/E denote Oracle/Estimated; FM/Diag denotes the specialized baseline.

| | Pure Low-Rank ($s=0$) | | | Pure Sparse ($r=0$) | | |
|---|---|---|---|---|---|---|
| $r$ or $s$ | SLR-O | SLR-E | FM-AG | SLR-O | SLR-E | Diag-AG |
| 1 | 1.71 | 1.71 | 1.66 | 1.19 | 1.19 | 0.76 |
| 5 | 1.87 | 1.88 | 1.82 | 1.55 | 1.55 | 1.28 |
| 10 | 1.96 | 1.97 | 1.91 | 1.70 | 1.70 | 1.51 |
| 20 | 1.94 | 1.95 | 1.87 | 1.90 | 1.90 | 1.77 |
| 30 | 2.07 | 2.08 | 2.00 | 2.08 | 2.08 | 1.97 |

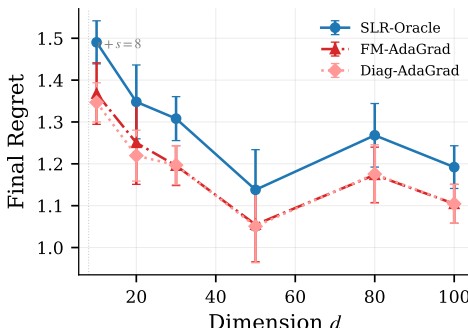

Figure A.2: Dimension scaling with fixed structure ($r=3$, $s=5$). Final regret remains approximately constant as $d$ grows from 10 to 100.

from which we obtain $-\langle \mathcal{G}_t, w^* \rangle \leq \Phi_t + \frac{1}{2}\|w^*\|^2_{H_t}$. Subtracting the same identity evaluated at times $t$ and $t-1$ yields the per-round bound

$$-\langle g_t, w^* \rangle \ \leq \ \Phi_t - \Phi_{t-1} + \tfrac{1}{2}\|w^*\|^2_{H_t - H_{t-1}}. \tag{17}$$

We next bound the learner's per-round loss. Since $w_t = -H_{t-1}^{-1}\mathcal{G}_{t-1}$, we have $\langle g_t, w_t \rangle = -g_t^\top H_{t-1}^{-1}\mathcal{G}_{t-1}$. Define $P := H_{t-1}^{-1} - H_t^{-1}$, which satisfies $P \succeq 0$ because $H_t \succeq H_{t-1}$ implies $H_t^{-1} \preceq H_{t-1}^{-1}$ by operator anti-monotonicity of the inverse on the PSD cone.

We now expand $\Phi_t - \Phi_{t-1}$ without dropping any terms. Writing $\mathcal{G}_t = \mathcal{G}_{t-1} + g_t$ and $H_t^{-1} = H_{t-1}^{-1} - P$:

$$\begin{aligned}
\Phi_t - \Phi_{t-1} &= \tfrac{1}{2}\mathcal{G}_t^\top H_t^{-1}\mathcal{G}_t - \tfrac{1}{2}\mathcal{G}_{t-1}^\top H_{t-1}^{-1}\mathcal{G}_{t-1} \\
&= \tfrac{1}{2}\mathcal{G}_{t-1}^\top (H_t^{-1} - H_{t-1}^{-1})\mathcal{G}_{t-1} + g_t^\top H_t^{-1}\mathcal{G}_{t-1} + \tfrac{1}{2}\|g_t\|^2_{H_t^{-1}} \\
&= -\tfrac{1}{2}\mathcal{G}_{t-1}^\top P\,\mathcal{G}_{t-1} + g_t^\top H_t^{-1}\mathcal{G}_{t-1} + \tfrac{1}{2}\|g_t\|^2_{H_t^{-1}}.
\end{aligned}$$

Adding $\langle g_t, w_t \rangle = -g_t^\top H_{t-1}^{-1}\mathcal{G}_{t-1} = -g_t^\top (H_t^{-1} + P)\mathcal{G}_{t-1}$ to the above:

$$\langle g_t, w_t \rangle + \Phi_t - \Phi_{t-1} \ = \ -g_t^\top P\,\mathcal{G}_{t-1} - \tfrac{1}{2}\mathcal{G}_{t-1}^\top P\,\mathcal{G}_{t-1} + \tfrac{1}{2}\|g_t\|^2_{H_t^{-1}}.$$

Completing the square in $\mathcal{G}_{t-1}$:

$$-g_t^\top P\,\mathcal{G}_{t-1} - \tfrac{1}{2}\mathcal{G}_{t-1}^\top P\,\mathcal{G}_{t-1} \ = \ -\tfrac{1}{2}(\mathcal{G}_{t-1} + g_t)^\top P\,(\mathcal{G}_{t-1} + g_t) + \tfrac{1}{2}g_t^\top P\,g_t \ = \ -\tfrac{1}{2}\mathcal{G}_t^\top P\,\mathcal{G}_t + \tfrac{1}{2}g_t^\top P\,g_t.$$

Since $P \succeq 0$, the term $\mathcal{G}_t^\top P\,\mathcal{G}_t \geq 0$ can be dropped, yielding

$$\langle g_t, w_t \rangle + \Phi_t - \Phi_{t-1} \ \leq \ \tfrac{1}{2}g_t^\top P\,g_t + \tfrac{1}{2}\|g_t\|^2_{H_t^{-1}} \ = \ \tfrac{1}{2}g_t^\top (P + H_t^{-1})g_t \ = \ \tfrac{1}{2}\|g_t\|^2_{H_{t-1}^{-1}},$$

where the last equality uses $P + H_t^{-1} = H_{t-1}^{-1}$ by definition of $P$.

Table A.2: Effect of decomposition noise $\epsilon$ on final regret ($d=50$, $T=1000$, $r=5$, $s=8$).

| $\epsilon$ | SLR-Oracle | SLR-Est | FM-AdaGrad | Diag-AdaGrad |
|---|---|---|---|---|
| 0.0 | $1.29 \pm 0.10$ | $1.28 \pm 0.10$ | $1.22 \pm 0.09$ | $1.21 \pm 0.09$ |
| 0.01 | $1.75 \pm 0.13$ | $1.74 \pm 0.13$ | $1.63 \pm 0.10$ | $1.62 \pm 0.11$ |
| 0.05 | $4.22 \pm 0.26$ | $4.14 \pm 0.24$ | $3.65 \pm 0.17$ | $3.65 \pm 0.19$ |
| 0.1 | $8.35 \pm 0.45$ | $8.04 \pm 0.38$ | $6.76 \pm 0.26$ | $6.75 \pm 0.30$ |
| 0.2 | $19.07 \pm 0.84$ | $17.67 \pm 0.65$ | $14.00 \pm 0.48$ | $13.97 \pm 0.59$ |
| 0.5 | $59.75 \pm 3.57$ | $52.12 \pm 3.43$ | $38.55 \pm 2.76$ | $38.36 \pm 3.07$ |

Combining the above with equation 17 gives the per-round regret bound

$$\langle g_t, w_t - w^* \rangle \ \leq \ \tfrac{1}{2}\|g_t\|^2_{H_{t-1}^{-1}} + \tfrac{1}{2}\|w^*\|^2_{H_t - H_{t-1}}.$$

Summing over $t = 1, \ldots, T$:

$$\sum_{t=1}^{T}\langle g_t, w_t - w^* \rangle \ \leq \ \tfrac{1}{2}\sum_{t=1}^{T}\|g_t\|^2_{H_{t-1}^{-1}} \ + \ \tfrac{1}{2}\sum_{t=1}^{T}\|w^*\|^2_{H_t - H_{t-1}}.$$

The second sum telescopes: $\sum_{t=1}^{T} w^{*\top}(H_t - H_{t-1})w^* = w^{*\top}(H_T - H_0)w^* \leq \|w^*\|^2_{H_T}$, completing the proof. □

## B.2 Proof of Lemma 2 (Trace-of-Square-Root Bound)

*Proof.* Define $A_t = \delta I + \sum_{\tau=1}^{t} v_\tau v_\tau^\top$ and $A_0 = \delta I$, so that $A_t = A_{t-1} + v_t v_t^\top$ and $H_t = A_t^{1/2}$.

The function $\phi(X) = \operatorname{tr}(X^{1/2})$ is concave on the positive definite cone $\mathbb{S}_{++}^d$. To see this, note that $X \mapsto X^{1/2}$ is operator concave (a classical result in matrix analysis), and the trace is a positive linear functional, so their composition is concave. The Fréchet derivative of $\phi$ at a point $X \in \mathbb{S}_{++}^d$ can be computed via spectral calculus: writing $X = \sum_i \lambda_i u_i u_i^\top$,

$$\frac{d}{d\varepsilon} \operatorname{tr}\big((X + \varepsilon\Delta)^{1/2}\big)\Big|_{\varepsilon=0} \ = \ \sum_i \frac{1}{2\lambda_i^{1/2}} u_i^\top \Delta\, u_i \ = \ \tfrac{1}{2}\operatorname{tr}(X^{-1/2}\Delta).$$

Applying the first-order concavity condition $\phi(Y) \leq \phi(X) + D\phi(X)[Y - X]$ at the point $X = A_t$ (the larger matrix), evaluated at $Y = A_{t-1} = A_t - v_t v_t^\top$:

$$\operatorname{tr}(A_{t-1}^{1/2}) \ \leq \ \operatorname{tr}(A_t^{1/2}) + D\phi(A_t)[A_{t-1} - A_t] \ = \ \operatorname{tr}(A_t^{1/2}) - \tfrac{1}{2}v_t^\top A_t^{-1/2} v_t.$$

Rearranging:

$$\tfrac{1}{2}\|v_t\|^2_{H_t^{-1}} \ = \ \tfrac{1}{2}v_t^\top A_t^{-1/2} v_t \ \leq \ \operatorname{tr}(A_t^{1/2}) - \operatorname{tr}(A_{t-1}^{1/2}).$$

Summing over $t = 1, \ldots, T$ and telescoping:

$$\sum_{t=1}^{T} \tfrac{1}{2}\|v_t\|^2_{H_t^{-1}} \ \leq \ \operatorname{tr}(A_T^{1/2}) - \operatorname{tr}(A_0^{1/2}) \ = \ \operatorname{tr}\left(\left(\delta I + \sum_{t=1}^{T} v_t v_t^\top\right)^{1/2}\right) - d\,\delta^{1/2}.$$

Multiplying both sides by 2 yields the first bound.

For the preconditioner-lag correction, define $P_t := H_{t-1}^{-1} - H_t^{-1} = A_{t-1}^{-1/2} - A_t^{-1/2}$, which satisfies $P_t \succeq 0$ by operator anti-monotonicity of $X \mapsto X^{-1/2}$. We decompose

$$\sum_{t=1}^{T}\|v_t\|^2_{H_{t-1}^{-1}} \ = \ \sum_{t=1}^{T}\|v_t\|^2_{H_t^{-1}} \ + \ \sum_{t=1}^{T} v_t^\top P_t v_t.$$

For each $t$, we bound $v_t^\top P_t v_t$ using the trace–operator-norm duality. Since $P_t \succeq 0$ and $v_t v_t^\top \succeq 0$:

$$v_t^\top P_t v_t \;=\; \operatorname{tr}(P_t\, v_t v_t^\top) \;\leq\; \operatorname{tr}(P_t)\, \|v_t v_t^\top\|_{\mathrm{op}} \;=\; \operatorname{tr}(P_t)\, \|v_t\|^2 \;\leq\; G^2 \operatorname{tr}(P_t),$$

where the first inequality is the standard bound $\operatorname{tr}(AB) \leq \operatorname{tr}(A)\,\|B\|_{\mathrm{op}}$ for PSD matrices $A, B$, and the last step uses $\|v_t\| \leq G$. Summing over $t$ and telescoping the trace:

$$\sum_{t=1}^{T} v_t^\top P_t v_t \;\leq\; G^2 \operatorname{tr}\left(\sum_{t=1}^{T} P_t\right) = G^2 \operatorname{tr}(A_0^{-1/2} - A_T^{-1/2}) \;\leq\; G^2 \operatorname{tr}(A_0^{-1/2}) \;=\; G^2 d\, \delta^{-1/2}.$$

Adding this to the first bound completes the proof. $\qquad\square$

## B.3   Proof of Lemma 3 (Diagonal AdaGrad Bound on Sparse Vectors)

*Proof.* Since $v_{t,j} = 0$ for all $j \notin \mathcal{J}$, the Mahalanobis norm reduces to a sum over the active support:

$$\|v_t\|_{D_t^{-1}}^2 \;=\; \sum_{j \in \mathcal{J}} \frac{v_{t,j}^2}{\left(\delta + \sum_{\tau=1}^{t} v_{\tau,j}^2\right)^{1/2}}.$$

For each coordinate $j \in \mathcal{J}$, define the accumulator $a_{t,j} = \delta + \sum_{\tau=1}^{t} v_{\tau,j}^2$, so that $a_{t,j} - a_{t-1,j} = v_{t,j}^2$.

By the algebraic identity $a^{1/2} - b^{1/2} = (a - b)\,/\,(a^{1/2} + b^{1/2}) \geq (a - b)\,/\,(2a^{1/2})$ valid for $a \geq b > 0$, we obtain

$$a_{t,j}^{1/2} - a_{t-1,j}^{1/2} \;\geq\; \frac{v_{t,j}^2}{2\,a_{t,j}^{1/2}}.$$

Rearranging gives $v_{t,j}^2 / a_{t,j}^{1/2} \leq 2(a_{t,j}^{1/2} - a_{t-1,j}^{1/2})$. Summing over all rounds $t = 1, \ldots, T$ and all coordinates $j \in \mathcal{J}$, the right-hand side telescopes:

$$\sum_{t=1}^{T} \|v_t\|_{D_t^{-1}}^2 \;\leq\; 2 \sum_{j \in \mathcal{J}} \left(a_{T,j}^{1/2} - a_{0,j}^{1/2}\right) \;=\; 2 \sum_{j \in \mathcal{J}} \left(\delta + \sum_{t=1}^{T} v_{t,j}^2\right)^{1/2} - 2|\mathcal{J}|\,\delta^{1/2},$$

since $a_{0,j} = \delta$ for all $j$. $\qquad\square$

## B.4   Proof of Lemma 4 (Trace Bound under Low-Rank Structure)

*Proof.* Let $U \in \mathbb{R}^{d \times r}$ be an orthonormal basis for the subspace, so that $L_t = U \hat{L}_t$ for some $\hat{L}_t \in \mathbb{R}^r$. Then

$$A \;=\; U\left(\sum_{t=1}^{T} \hat{L}_t \hat{L}_t^\top\right) U^\top,$$

and the eigenvalues of $A$ are the $r$ eigenvalues $\lambda_1 \geq \cdots \geq \lambda_r \geq 0$ of the $r \times r$ matrix $\hat{A} := \sum_t \hat{L}_t \hat{L}_t^\top$, together with $(d - r)$ zero eigenvalues (since $A$ has rank at most $r$).

Expressing the trace via the eigendecomposition:

$$\operatorname{tr}\left((\delta I + A)^{1/2}\right) \;=\; \sum_{i=1}^{r} (\delta + \lambda_i)^{1/2} + (d - r)\,\delta^{1/2}.$$

Since $\lambda_i \leq \lambda_1 = \sigma_1(A)$ for all $i \in [r]$, each term in the first sum satisfies $(\delta + \lambda_i)^{1/2} \leq (\delta + \lambda_1)^{1/2}$, giving

$$\sum_{i=1}^{r} (\delta + \lambda_i)^{1/2} \;\leq\; r\,(\delta + \sigma_1(A))^{1/2},$$

which yields equation 8.

For the Cauchy–Schwarz bound equation 9, write $\mathrm{tr}(A^{1/2}) = \sum_{i=1}^{R} \lambda_i^{1/2}$ and apply the Cauchy–Schwarz inequality with the all-ones vector:

$$\left(\sum_{i=1}^{R} 1 \cdot \lambda_i^{1/2}\right)^2 \;\leq\; R \cdot \sum_{i=1}^{R} \lambda_i \;=\; R \cdot \mathrm{tr}(A),$$

and $\mathrm{tr}(A) = \mathrm{tr}(\sum_t L_t L_t^\top) = \sum_t \|L_t\|_2^2$. □

### B.5  Proof of Lemma 5 (Coin-Betting Meta-Algorithm Regret)

*Proof.* By linearity of the inner product,

$$\langle g_t, w_t \rangle \;=\; \alpha_t \langle g_t, w_t^L \rangle + (1 - \alpha_t)\langle g_t, w_t^S \rangle.$$

Define the per-round difference $\ell_t := \langle g_t, w_t^L - w_t^S \rangle$ and the normalized coin $c_t := \ell_t/L$. By Cauchy–Schwarz and Assumptions 2–6, $|\ell_t| \leq \|g_t\| \, \|w_t^L - w_t^S\| \leq G \cdot 2B_w = L$, so $|c_t| \leq 1$.

The meta-problem reduces to one-dimensional online linear optimization on $[0,1]$ with per-round loss $\alpha \mapsto \alpha \ell_t$. By the Orabona & Pál (2016) reduction of OCO on $[0,1]$ to coin-betting, the KT strategy produces $\alpha_t \in [0,1]$ satisfying, for any $\alpha^* \in [0,1]$:

$$\sum_{t=1}^{T} (\alpha_t - \alpha^*)\, c_t \;\leq\; |\alpha^*| \sqrt{2\sum_t c_t^2 \cdot \ln(T+1)} + O(\ln T).$$

Since $L$ is a *fixed* constant (not time-varying), the un-normalization is an exact algebraic identity:

$$\sum_{t=1}^{T} (\alpha_t - \alpha^*)\, \ell_t \;=\; L \cdot \sum_{t=1}^{T} (\alpha_t - \alpha^*)\, c_t \;\leq\; |\alpha^*| \sqrt{2V_T \ln(T+1)} + O(L \ln T),$$

where $V_T = \sum_t \ell_t^2 = L^2 \sum_t c_t^2$.

To assemble the final bound, write

$$\sum_{t=1}^{T} \langle g_t, w_t \rangle \;=\; \sum_{t=1}^{T} \langle g_t, w_t^S \rangle + \sum_{t=1}^{T} \alpha_t \ell_t \;=\; \sum_{t=1}^{T} \langle g_t, w_t^S \rangle + \alpha^* \sum_{t=1}^{T} \ell_t + \sum_{t=1}^{T} (\alpha_t - \alpha^*)\, \ell_t.$$

Noting that $\sum_t \langle g_t, w_t^S \rangle + \alpha^* \sum_t \ell_t = \alpha^* \sum_t \langle g_t, w_t^L \rangle + (1 - \alpha^*)\sum_t \langle g_t, w_t^S \rangle$ and using $|\alpha^*| \leq 1$ gives equation 10. □

### B.6  Proof of Theorem 1 (Main Result)

*Proof.* By Assumption 1 and the linearized regret inequality equation 2,

$$\mathrm{Regret}_T(w^*) \;\leq\; \sum_{t=1}^{T} \langle g_t, \, w_t - w^* \rangle.$$

Applying Lemma 5 (valid under Assumptions 2 and 6), for any $\alpha^* \in [0,1]$:

$$\mathrm{Regret}_T(w^*) \;\leq\; \alpha^* \sum_{t=1}^{T} \langle g_t, w_t^L - w^* \rangle + (1-\alpha^*)\sum_{t=1}^{T} \langle g_t, w_t^S - w^* \rangle + \sqrt{2V_T \ln(T+1)} + O(GB_w \ln T). \quad (18)$$

It remains to bound the regret of each sub-algorithm.

Consider the low-rank sub-algorithm $\mathcal{A}^L$, which is FTRL with preconditioner $H_t = M_t = (\delta I + \sum_{\tau \le t} \hat{L}_\tau \hat{L}_\tau^\top)^{1/2}$, receiving the full gradient $g_t$ as its loss. By Lemma 1:

$$R_T^L := \sum_{t=1}^T \langle g_t, w_t^L - w^* \rangle \le \tfrac{1}{2}\|w^*\|_{M_T}^2 + \tfrac{1}{2}\sum_{t=1}^T \|g_t\|_{M_{t-1}^{-1}}^2.$$

To bound the stability term $\sum_t \|g_t\|_{M_{t-1}^{-1}}^2$, write $g_t = \hat{L}_t + (\hat{S}_t + E_t)$ where $E_t = g_t - \hat{S}_t - \hat{L}_t$ is the decomposition error, and use the inequality $\|a+b\|_P^2 \le 2\|a\|_P^2 + 2\|b\|_P^2$ (valid for any PSD $P$) to obtain

$$\|g_t\|_{M_{t-1}^{-1}}^2 \le 2\|\hat{L}_t\|_{M_{t-1}^{-1}}^2 + 2\|\hat{S}_t + E_t\|_{M_{t-1}^{-1}}^2.$$

The first term captures the "matched" component, i.e., the low-rank gradient measured by the low-rank preconditioner. Lemma 2 (with preconditioner-lag correction equation 6) gives

$$\sum_{t=1}^T \|\hat{L}_t\|_{M_{t-1}^{-1}}^2 \le 2\operatorname{tr}\big((\delta I + \hat{\mathbf{G}}_T^L)^{1/2}\big) - 2d\,\delta^{1/2} + G^2 d\,\delta^{-1/2}.$$

The second term captures the "cross-contamination," i.e., the sparse-plus-error component measured by the low-rank preconditioner. Since $A_{t-1}^L = \delta I + \sum_{\tau < t} \hat{L}_\tau \hat{L}_\tau^\top \succeq \delta I$, we have $M_{t-1} = (A_{t-1}^L)^{1/2} \succeq \delta^{1/2} I$, and hence $M_{t-1}^{-1} \preceq \delta^{-1/2} I$. Therefore,

$$\|\hat{S}_t + E_t\|_{M_{t-1}^{-1}}^2 \le \delta^{-1/2}\|\hat{S}_t + E_t\|^2 \le 2\delta^{-1/2}\big(\|\hat{S}_t\|^2 + \|E_t\|^2\big),$$

and summing over $t$:

$$\sum_{t=1}^T \|\hat{S}_t + E_t\|_{M_{t-1}^{-1}}^2 \le \frac{2}{\delta^{1/2}}\Big(\sum_{t=1}^T \|\hat{S}_t\|^2 + \mathcal{E}_T^2\Big).$$

Combining these estimates (applying the factor $\tfrac{1}{2}$ from Lemma 1 to the two terms, each carrying the factor 2 from the triangle inequality) yields equation 12.

An analogous argument bounds the sparse sub-algorithm $\mathcal{A}^S$, which is FTRL with diagonal preconditioner $H_t = D_t$, also receiving the full gradient $g_t$. By Lemma 1:

$$R_T^S := \sum_{t=1}^T \langle g_t, w_t^S - w^* \rangle \le \tfrac{1}{2}\|w^*\|_{D_T}^2 + \tfrac{1}{2}\sum_{t=1}^T \|g_t\|_{D_{t-1}^{-1}}^2.$$

Writing $g_t = \hat{S}_t + (\hat{L}_t + E_t)$ and applying the same decomposition:

$$\|g_t\|_{D_{t-1}^{-1}}^2 \le 2\|\hat{S}_t\|_{D_{t-1}^{-1}}^2 + 2\|\hat{L}_t + E_t\|_{D_{t-1}^{-1}}^2.$$

The diagonal analogue of Lemma 2 (obtained by the same concavity argument applied coordinate-wise, with lag correction $G^2 d\,\delta^{-1/2}$) combined with Lemma 3 gives

$$\sum_{t=1}^T \|\hat{S}_t\|_{D_{t-1}^{-1}}^2 \le 2\sum_{j \in \mathcal{J}} \big(\delta + (\hat{\mathbf{G}}_T^S)_{jj}\big)^{1/2} - 2|\mathcal{J}|\,\delta^{1/2} + G^2 d\,\delta^{-1/2}.$$

Since $D_{t-1} \succeq \delta^{1/2} I$, the cross-contamination term satisfies

$$\sum_{t=1}^T \|\hat{L}_t + E_t\|_{D_{t-1}^{-1}}^2 \le \frac{2}{\delta^{1/2}}\Big(\sum_{t=1}^T \|\hat{L}_t\|^2 + \mathcal{E}_T^2\Big).$$

Combining yields equation 13.

Substituting the bounds equation 12 and equation 13 into the meta-algorithm decomposition equation 18 and taking the minimum over $\alpha^* \in [0, 1]$ yields

$$\text{Regret}_T(w^*) \leq \min_{\alpha^* \in [0,1]} \{\alpha^* R_T^L + (1 - \alpha^*) R_T^S\} + \sqrt{2V_T \ln(T+1)} + O(GB_w \ln T),$$

which is equation 11.

We record explicit bounds on the cross-contamination terms for later use. By the definition of Algorithm 1, $\hat{L}_t = \text{Proj}_r(g_t; U_{t-1})$ is an orthogonal projection, so $\|\hat{L}_t\| \leq \|g_t\| \leq G$ by Assumption 2, giving $\sum_t \|\hat{L}_t\|^2 \leq G^2 T$. For $\hat{S}_t = \text{Hard}_s(g_t - \hat{L}_t)$, hard thresholding is a contraction in Euclidean norm, so $\|\hat{S}_t\| \leq \|g_t - \hat{L}_t\| \leq \|g_t\| + \|\hat{L}_t\| \leq 2G$, giving $\sum_t \|\hat{S}_t\|^2 \leq 4G^2 T$. Therefore, the cross-contamination contributions in $R_T^L$ and $R_T^S$ are bounded by $O(G^2 T/\delta^{1/2})$, and the preconditioner-lag corrections contribute $O(G^2 d/\delta^{1/2})$ each. $\qquad \square$

### B.7 Proof of Theorem 2 (Per-Term Lower Bound)

*Proof.* We construct two separate adversaries and lower-bound the regret against the worse of the two.

Consider first the purely low-rank adversary. Set $S_t = 0$ and $L_t = g_t$, with all gradients lying in the fixed subspace $V = \text{span}(e_1, \ldots, e_r)$. Concretely, for each round $t$ and each coordinate $j \in \{1, \ldots, r\}$, draw $g_{t,j} \in \{+G/\sqrt{r}, -G/\sqrt{r}\}$ independently and uniformly (i.i.d. Rademacher), and set $g_{t,j} = 0$ for $j > r$. By construction, $\|g_t\|^2 = r \cdot G^2/r = G^2$, so Assumption 2 is satisfied.

The gradient outer-product matrix is $\mathbf{G}_T^L = \sum_t g_t g_t^\top$. Since the entries $g_{t,j}$ are i.i.d. with $\mathbb{E}[g_{t,j}^2] = G^2/r$, by the law of large numbers all $r$ eigenvalues of $\mathbf{G}_T^L$ concentrate around $G^2 T/r$, so $\sigma_1(\mathbf{G}_T^L) = G^2 T/r$ in expectation.

By the standard minimax lower bound for online linear optimization on $\mathbb{R}^r$ with bounded gradients (Cutkosky et al., 2023; Duchi et al., 2011), which follows from the minimax theorem applied to the zero-sum game between the learner and the Rademacher adversary, for any online algorithm there exists a gradient sequence such that

$$\mathbb{E}[\text{Regret}_T] \geq \tfrac{1}{2}\|w^*\| G\sqrt{T}.$$

Substituting $G\sqrt{T} = \sqrt{r \cdot (G^2 T/r)} = \sqrt{r \cdot \sigma_1(\mathbf{G}_T^L)}$:

$$\mathbb{E}[\text{Regret}_T] \geq \tfrac{1}{2}\|w^*\|\sqrt{r \cdot \sigma_1(\mathbf{G}_T^L)}.$$

Consider next the purely sparse adversary. Set $L_t = 0$ and $S_t = g_t$, with $s$-sparse Rademacher gradients supported on coordinates $\{1, \ldots, s\}$: $g_{t,j} \in \{+G/\sqrt{s}, -G/\sqrt{s}\}$ i.i.d. Rademacher for $j \leq s$, and $g_{t,j} = 0$ for $j > s$. Again $\|g_t\|^2 = G^2$, and each coordinate accumulates $\sum_t g_{t,j}^2 = G^2 T/s$, so $\|\mathbf{G}_T^S\|_\infty = \max_j \sum_t g_{t,j}^2 = G^2 T/s$.

Choose the comparator $w^*$ with $|w_j^*| = \|w^*\|/\sqrt{s}$ for $j \leq s$ and $w_j^* = 0$ otherwise (so that $\|w^*\|^2 = s \cdot \|w^*\|^2/s = \|w^*\|^2$). Since the problem decomposes into $s$ independent one-dimensional sub-problems, the standard one-dimensional minimax lower bound $\Omega(|w_j^*|\sqrt{\sum_t g_{t,j}^2})$ applies to each coordinate. Summing over coordinates:

$$\mathbb{E}[\text{Regret}_T] \geq \sum_{j=1}^s \tfrac{1}{2}|w_j^*|\sqrt{\sum_{t=1}^T g_{t,j}^2} = \sum_{j=1}^s \tfrac{1}{2}\frac{\|w^*\|}{\sqrt{s}} \cdot \frac{G\sqrt{T}}{\sqrt{s}} = \tfrac{1}{2}\|w^*\| G\sqrt{T}.$$

Substituting $G\sqrt{T} = \sqrt{s \cdot (G^2 T/s)} = \sqrt{s \cdot \|\mathbf{G}_T^S\|_\infty}$:

$$\mathbb{E}[\text{Regret}_T] \geq \tfrac{1}{2}\|w^*\|\sqrt{s \cdot \|\mathbf{G}_T^S\|_\infty}.$$

Since any algorithm must face both adversaries, its worst-case expected regret is at least

$$\max\left(\tfrac{1}{2}\|w^*\|\sqrt{r \, \sigma_1(\mathbf{G}_T^L)}, \ \tfrac{1}{2}\|w^*\|\sqrt{s \, \|\mathbf{G}_T^S\|_\infty}\right).$$

Applying the elementary inequality $\max(a,b) \geq (a+b)/2$ with $a = \frac{1}{2}\|w^*\|\sqrt{r\sigma_1}$ and $b = \frac{1}{2}\|w^*\|\sqrt{s\|\mathbf{G}_T^S\|_\infty}$ yields

$$\max(a,b) \;\geq\; \tfrac{1}{4}\,\|w^*\|\Big(\sqrt{r\,\sigma_1(\mathbf{G}_T^L)} + \sqrt{s\,\|\mathbf{G}_T^S\|_\infty}\Big),$$

which is the claimed bound. $\qquad\square$

### B.8 Proof of Corollary 3 (Pure-Structure Special Cases)

*Proof.* Both cases follow by direct substitution into Theorem 1 with the indicated choice of $\alpha^*$.

For part (a), when $r = 0$ we have $\hat{L}_t = 0$ and $\hat{S}_t = g_t$ for all $t$, so $E_t = 0$ and $\mathcal{E}_T = 0$. Setting $\alpha^* = 0$ selects the sparse sub-algorithm exclusively, and the low-rank sub-algorithm's regret $R_T^L$ drops out of the bound. Within $R_T^S$, the cross-contamination term $\sum_t \|\hat{L}_t\|^2 = 0$ vanishes since $\hat{L}_t = 0$, yielding equation 15. The remaining terms (the comparator penalty $\|w^*\|_{D_T}^2$, the stability sum $\sum_{j\in\mathcal{J}}(\delta + (\hat{\mathbf{G}}_T^S)_{jj})^{1/2}$, and the preconditioner-lag correction $G^2 d\,\delta^{-1/2}$) are precisely the standard diagonal AdaGrad bound.

For part (b), the argument is symmetric: when $s = 0$ we have $\hat{S}_t = 0$ and $\hat{L}_t = g_t$, and setting $\alpha^* = 1$ selects the low-rank sub-algorithm. The cross-contamination term $\sum_t \|\hat{S}_t\|^2 = 0$ vanishes, yielding equation 16. Under stable rank-$r$ structure, Lemma 4 further simplifies the trace term $\mathrm{tr}((\delta I + \hat{\mathbf{G}}_T^L)^{1/2})$ to $r(\delta + \sigma_1(\hat{\mathbf{G}}_T^L))^{1/2} + (d-r)\delta^{1/2}$, recovering the full-matrix AdaGrad bound specialized to rank-$r$ gradients. $\quad\square$

### B.9 Proof of Corollary 4 (Adaptive Interpolation)

*Proof.* By Lemma 5 with the fixed normalization $L = 2GB_w$, the variance proxy satisfies

$$V_T \;=\; \sum_{t=1}^{T}\langle g_t,\, w_t^L - w_t^S\rangle^2 \;\leq\; L^2 T \;=\; 4G^2 B_w^2 T.$$

Therefore,

$$\sqrt{2V_T\ln(T{+}1)} \;\leq\; 2\,GB_w\,\sqrt{2T\ln(T{+}1)} \;=\; \widetilde{O}(GB_w\sqrt{T}).$$

The un-normalization in the proof of Lemma 5 is an exact algebraic identity (since $L$ is a fixed constant, not time-varying), so no approximation gap is introduced. $\qquad\square$

## C Discussion of Assumption 6 in Unconstrained Settings

Assumption 6 requires a known constant $B_w$ such that $\|w_t^S\|_2, \|w_t^L\|_2 \leq B_w$ for all $t$. It is used in exactly one place in our analysis, Lemma 5, where it ensures that the scaled coin outcome $c_t = \langle g_t, w_t^L - w_t^S\rangle/(2GB_w)$ lies in $[-1, 1]$, so that the Krichevsky–Trofimov coin-betting reduction applies with a fixed normalization constant. When $\mathcal{W}$ is bounded, the assumption is automatic with $B_w$ equal to the domain diameter; in the unconstrained setting $\mathcal{W} = \mathbb{R}^d$, three concrete ways to address it are available. First, the FTRL updates $w_{t+1}^S = -D_t^{-1}\Sigma_t$ and $w_{t+1}^L = -M_t^{-1}\Sigma_t$, together with $D_t, M_t \succeq \delta^{1/2}I$ (from the initialization at $\delta\mathbf{1}_d$ and $\delta I$), give the deterministic a priori bound $\|w_{t+1}^L\|_2, \|w_{t+1}^S\|_2 \leq \delta^{-1/2}Gt$, so that $B_w := \delta^{-1/2}GT$ always satisfies the assumption, although substituting this naive bound into Lemma 5 produces a super-linear $\widetilde{O}(\delta^{-1/2}G^2T^{3/2})$ meta-overhead and is therefore only useful when a problem-dependent constant bound on $\|\Sigma_t\|$ is available (e.g., for strongly convex or self-concordant losses). Second, applying a doubling trick to the normalization $L = 2GB_w$, with $\hat{L}_k = 2G\cdot2^k$ and a restart of the meta-algorithm whenever $|\langle g_t, w_t^L - w_t^S\rangle| > \hat{L}_k$, removes any a priori knowledge of $B_w$; the number of restarts is at most $\lceil\log_2(B_w^\star/B_w^{(0)})\rceil$, and the meta-overhead becomes $\widetilde{O}(GB_w^\star\sqrt{T})$ with the data-dependent supremum $B_w^\star := \sup_t \max(\|w_t^L\|, \|w_t^S\|)$ replacing the a priori $B_w$. Third, replacing the KT meta-algorithm with a comparator-norm-adaptive coin-betting strategy in the style of Cutkosky & Orabona (2018) or Mhammedi & Koolen (2020) yields regret $\widetilde{O}(\|\alpha^*\|\sqrt{\sum_t \ell_t^2}) \leq \widetilde{O}(GB_w^\star\sqrt{T})$ where $\ell_t = \langle g_t, w_t^L - w_t^S\rangle$, without requiring any a priori bound on $|\ell_t|$, and leaves the structural terms $R_T^L, R_T^S$ in Theorem 1 unaffected.

To summarize, the role of Assumption 6 is twofold: at the algorithm level it provides a single constant $B_w$ for the fixed normalization $L = 2GB_w$, and at the regret-bound level it renders the meta-overhead $\widetilde{O}(GB_w\sqrt{T})$ a finite quantity. The doubling-trick and norm-adaptive variants remove the algorithm-level dependence on $B_w$, so SLR-FTRL can be implemented in a fully parameter-free fashion in $B_w$, but the regret still depends on the data-dependent supremum $B_w^\star$; this dependence cannot be removed without further problem-dependent assumptions, since the meta-regret intrinsically scales with the divergence between the two sub-algorithms' iterates. On benign sequences with $B_w^\star = O(1)$, the meta-overhead is $\widetilde{O}(\sqrt{T})$ as quoted alongside Theorem 1; on pathological sequences with $B_w^\star = \Theta(T)$, it degrades gracefully to $\widetilde{O}(T^{3/2})$. We adopt the bounded-iterate version in the main text purely to keep the exposition self-contained.

## D  Approximation Methods for Higher-Dimensional and Higher-Rank Workloads

The per-round cost of Algorithm 1, $O(dr^2 + d)$ time and $O(dr)$ memory, is dominated by maintaining the rank-$r$ subspace basis $U_t$ and the low-rank eigendecomposition of $A_t^L = \delta I + \sum_{\tau \le t} \hat{L}_\tau \hat{L}_\tau^\top$. For large-scale workloads where $d$ is in the millions and $r$ in the hundreds, several approximation strategies preserve the structural form of our regret bound up to an additive sketch-error term. A streaming randomized SVD (Halko et al., 2011) of rank $r+p$ with oversampling $p = O(\log r)$ drops the per-round cost to $O(dr)$ and introduces an additive term proportional to $\sqrt{T \cdot \sigma_{r+1}(\mathbf{G}_T^L)}/\delta^{1/2}$ that is negligible when $\sigma_{r+1} \ll \sigma_1$. A Frequent-Directions sketch (Ghashami et al., 2016) of width $\ell \ge r$ runs in $O(d\ell)$ time and memory and preserves the trace bound in Lemma 2 up to a multiplicative $(1 + \epsilon_{\mathrm{FD}})$ factor with $\epsilon_{\mathrm{FD}} = O(\mathrm{tr}(A_t^L)/((\ell - r)\sigma_1))$, giving a smooth interpolation between the diagonal ($\ell = 0$) and full-matrix ($\ell = d$) regimes. For deep-learning workloads organized as a sequence of weight tensors $W^{(1)}, \ldots, W^{(K)}$, the low-rank preconditioner $M_t$ can be replaced by a block-diagonal Kronecker-factored preconditioner with one block per layer in the style of Shampoo (Gupta et al., 2018), reducing the per-layer cost from $O((mn)r^2)$ to $O((m^2 + n^2)r)$ at the price of a Kronecker approximation factor bounded under standard layer-wise gradient assumptions. Updating $M_t$ only every $k$ rounds (typically $k = 10$ to $100$) reduces the amortized per-round cost from $O(dr^2)$ to $O(dr^2/k + dr)$ at the price of an additional cross-contamination term $O(k \cdot G^2/\delta^{1/2})$ per interval, and choosing $k = \Theta(\sqrt{T})$ gives an additive $\widetilde{O}(\sqrt{T})$ correction.

These four strategies are largely independent and can be combined, for example by using Kronecker-factored blocks together with Frequent-Directions sketching per block, refreshed every $k$ rounds. As a practical recommendation, we suggest the exact algorithm for $d \le 10^4$ and $r \le 100$; Frequent-Directions sketching with $\ell = 2r$ combined with lazy updates every $k = O(\sqrt{T})$ rounds for $d \in [10^4, 10^6]$; and Kronecker-factored blocks with per-block Frequent-Directions sketching for $d > 10^6$, which has the additional benefit of preserving the natural parameter layout of the model. A full empirical study of these approximations on production-scale neural-network training is an important direction for future work.

