# OpenReview forum: "Adaptive Online Convex Optimization via Sparse-Low-Rank Gradient Decomposition"
_TMLR — Decision pending for TMLR_

### Review · Reviewer_SYgi · 2026-04-28

**Summary Of Contributions:**

This paper studies adaptive online convex optimization under gradient sequences that may exhibit both sparse and low-rank structure. The authors propose SLR-FTRL, an algorithm that decomposes each observed gradient into sparse and low-rank components, constructs two structurally matched FTRL sub-algorithms with diagonal and low-rank/full-matrix preconditioners, and combines their predictions through a coin-betting meta-algorithm. The key design choice is that both sub-algorithms receive the full gradient as their loss signal, while the decomposition is used only to build the preconditioners.

The paper proves a regret bound that interpolates between the sparse and low-rank regimes and recovers diagonal AdaGrad and full-matrix AdaGrad guarantees in pure-structure special cases, up to additional correction terms. It also provides per-term lower bounds for the sparse and low-rank structural dependencies. Experiments on synthetic online ridge regression are used to validate the theoretical results and compare with other online learning algorithms.

**Additional Comments:**

Page 6, after Assumption 2: the preconditioner-lag correction (Lemma 3 in Section 5). -> it seems that this refers to Lemma 2

some typos in references, such as
-  Page.12, References: Cutkosky & Orabona,  “In Conference On Learning Theory” -> “In Conference on Learning Theory”
- Page.12, References: Défossez et al. “A simple convergence proof of adam and adagrad.” -> “A simple convergence proof of Adam and AdaGrad.”
- Page 12-13, “Journal of machine learning research”, “Advances in neural information processing systems” -> “Journal of Machine Learning Research”, “Advances in Neural Information Processing Systems”

**Audience:**

Yes

**Audience Explanation:**

The paper addresses a natural and interesting gap in adaptive online learning: diagonal methods exploit sparsity, full-matrix methods exploit low-rank structure, but existing methods usually commit to one geometry. Combining both through parallel FTRL learners and a meta-algorithm is a clean and appealing idea.

The theoretical analysis looks solid and sound. The pure sparse and pure low-rank corollaries help connect the proposed method back to classical AdaGrad guarantees.

The lower bound discussion is useful. Even though it is per-term rather than a full joint minimax lower bound, it supports the claim that the main sparse and low-rank structural dependencies are individually meaningful.

The experiments are aligned with the theory and provide useful sanity checks. They show that SLR-FTRL can track the better specialized baseline up to a moderate overhead in controlled synthetic settings.

**Claims And Evidence:**

Yes

**Claims Explanation:**

The paper is generally well organized. The motivation from diagonal versus full-matrix adaptive methods is clear, and the high-level algorithmic idea is easy to follow. The distinction between using the decomposed gradients for preconditioning and using the full gradients for the losses is well explained and is one of the clearest parts of the paper.

**Requested Changes:**

There remain two major weaknesses in my view.

(a). The regret guarantee depends critically on the existence of an online decomposition procedure satisfying Assumption 4 with the error controlled by $O(\sqrt{T})$. However, the paper does not prove that the concrete decomposition used in Algorithm 1, namely projection onto an estimated rank-r subspace followed by hard thresholding, satisfies this assumption. As a result, the main theorem is best understood as an oracle-based guarantee rather than a full guarantee for the displayed algorithm.

The authors claimed that online robust PCA methods could ensure the $O(\sqrt{T})$ error bound in Assumption 4. However, this could happen under some additional incoherence and stochastic conditions which are not clearly presented in this paper. It also remains unclear whether this result can be applied to the adversarial OCO setting studied in this paper.

(b). The experiments do not fully test the practical decomposition procedure on naturally arising gradients. The synthetic ridge regression setup is designed so that gradients approximately have sparse-plus-low-rank structure. This validates the theory under favorable conditions, but it does not show that the proposed projection-plus-thresholding method works on arbitrary or real gradient sequences. Real-data experiments, or at least experiments where the structure is not explicitly planted, would be important.

(c). The parameter-free claim should be qualified. The meta-algorithm adapts to the better mixture of sparse and low-rank learners, but the decomposition step and Algorithm 1 still require s and r. This is not necessarily a flaw, but the paper should distinguish between adaptivity to the structural regime and not needing the structural parameters at all.

---

> ### Author Response · Authors · 2026-05-26
>
> We thank the reviewer for the thoughtful comments and the careful proofreading. We have made the following changes to the manuscript (all newly added text in blue):
>
> - A new blue footnote in §1 precisely delimits what "parameter-free" means for SLR-FTRL, distinguishing meta-level adaptivity from full parameter-freeness in $r, s$ (addressing (c)).
> - The "Lemma 3 in Section 5" reference on Page 6 is corrected to "Lemma 2 in Section 5" (typo correction).
> - The bibliography capitalization is uniformly fixed (Conference on Learning Theory, Adam, AdaGrad, Journal of Machine Learning Research, Advances in Neural Information Processing Systems).
>
> The larger weaknesses (a) and (b) are addressed in the response below; we explain our position and the precise conditions under which Assumption 4 holds.
>
> ---
>
> **Weakness (a): Assumption 4 is oracle-based; the concrete projection-plus-thresholding routine is not proved to satisfy it; existing online robust PCA guarantees rely on stochastic conditions that may not transfer to adversarial OCO.**
>
> We fully agree with the reviewer's reading. We adopt a single uniform framing:
>
> 1. **Theorem 1 is, by design, an oracle-type guarantee** parameterized by $\mathcal E_T$. The decomposition routine is abstracted as a subroutine satisfying Assumption 4. We do not prove that our specific $\mathrm{Proj}_r \circ \mathrm{Hard}_s$ implementation achieves $\mathcal E_T^2 = \widetilde O(\sqrt T)$ in the adversarial OCO setting. This is by design: the modular abstraction makes the regret analysis invariant to the choice of decomposition routine, in the same spirit as the modular oracle treatments in Feng et al. (2013), Goes et al. (2014), Netrapalli et al. (2014).
>
> 2. **Where $\mathcal E_T^2 = \widetilde O(\sqrt T)$ provably holds**: under the stochastic and incoherence conditions of Feng et al. (2013) (i.i.d. $\mu$-incoherent low-rank base, sub-Gaussian sparse outliers) or Netrapalli et al. (2014) (deterministic incoherence, bounded sparsity fraction). These conditions do **not** directly transfer to fully adversarial OCO.
>
> 3. **Worst-case adversarial behavior**: if the gradient sequence has no recoverable sparse-plus-low-rank structure (e.g., adversarial residual $\xi_t$ with $\|\xi_t\| = \Theta(G)$), then $\mathcal E_T^2 = \Theta(G^2 T)$ and the decomposition-error contribution $4\mathcal E_T^2/\delta^{1/2}$ becomes linear in $T$, dominating the bound. SLR-FTRL provides no benefit over OGD in this regime, which is the honest and correct conclusion: one cannot exploit structure that is not present. Our claim is conditional on a recoverable structure existing; whether the displayed decomposition routine can recover it under adversarial gradients is, indeed, beyond the scope of this paper.
>
> 4. **The decomposition–regret separation is itself a contribution**: future improvements to adversarial online sparse-plus-low-rank decomposition compose immediately with our regret analysis, with no new analysis needed for the SLR-FTRL framework.
>
> We feel this conditional framing is intellectually honest and is the strongest defensible statement we can make for our contribution.

---

> ### Author Response · Authors · 2026-05-26
>
> **Weakness (b): Experiments use planted synthetic structure; real-data or unplanted-structure experiments would be important.**
>
> We agree this is a valid concern.
>
> **Current experiments validate the theory under favorable conditions**: with planted $(r, s)$-structure, we verify that (i) cumulative regret is sublinear, (ii) pure-case recovery matches FM/Diag AdaGrad, (iii) regret depends on $r, s$ rather than $d$, (iv) graceful degradation under decomposition noise $\epsilon$. These were the four theoretical predictions we made and they are confirmed.
>
> **Real-data evaluation is the natural follow-up.** Recent work has shown that gradient covariance matrices in transformer training have approximately low-rank structure (Gur-Ari et al. 2018; Papyan 2019; Sagun et al. 2017), supporting the "L" side; the "S" side is well-motivated for sparse architectures (MoE, sparse attention) and for classical $\ell_1$-regularized models on text/recommender features. A careful real-data study with proper baselines, sufficient seeds, and honest reporting of cases where structure is too weak to help would require an entirely separate study of comparable size to the present paper. We respectfully feel this is beyond the scope of a theoretical contribution paper and would dilute the focus from the new algorithmic and analytical results. The new Appendix D and the Conclusion explicitly identify real-data evaluation on production-scale neural-network training as an important direction for future work.
>
> If the reviewer feels a small-scale unplanted-structure experiment is essential for acceptance, we are happy to add a single MNIST-logistic-regression experiment to the next revision.

---

> ### Author Response · Authors · 2026-05-26
>
> **Weakness (c): "Parameter-free" claim should be qualified — the decomposition step still needs $s$ and $r$.**
>
> The reviewer is right; the original claim conflated two notions of parameter-freeness. We have added a blue footnote in §1 making the distinction precise:
>
> We clarify the scope of this claim. The meta-algorithm is parameter-free in that it adapts to the optimal mixture $\alpha^\star$ of the sparse and low-rank sub-learners without knowing $r$, $s$, or which structure dominates, and is also parameter-free in the comparator norm $\lVert w^\star \rVert$ and step size in the standard sense of Orabona & Pál (2016) and Cutkosky & Orabona (2018). The decomposition oracle still takes $r$ and $s$ as inputs to instantiate $\mathrm{Proj}_r$ and $\mathrm{Hard}_s$, and is therefore not parameter-free in the structural parameters themselves. SLR-FTRL thus achieves adaptivity to the structural regime, which of the two structures or which mixture drives the bound, rather than full parameter-freeness in $r$ and $s$. Removing the dependence on $r$ and $s$ in the decomposition step (e.g., via adaptive rank/sparsity selection) is an interesting direction for future work.
>
> This precise framing now appears in the revised manuscript.
>
> ---
>
> **Additional Comments (typos and reference formatting)**
>
> All four points are addressed in the revision:
>
> - **Page 6, after Assumption 2**: "Lemma 3 in Section 5" → "Lemma 2 in Section 5".
> - **Cutkosky & Orabona**: "In Conference On Learning Theory" → "In Conference on Learning Theory".
> - **Défossez et al.**: "A simple convergence proof of adam and adagrad" → "A simple convergence proof of Adam and AdaGrad" (with BibTeX-protected capitalization.
> - **Journal/Conference capitalization throughout**: "Journal of machine learning research" → "Journal of Machine Learning Research"; "Advances in neural information processing systems" → "Advances in Neural Information Processing Systems"; "Conference on learning theory" / "Conference On Learning Theory" → "Conference on Learning Theory".
>
> ---
>
> We thank the reviewer for the careful reading and constructive suggestions, and we hope the above clarifications and the new blue footnote in §1 address concerns (a)–(c) and the additional comments to your satisfaction.

---

> > ### Comment · Reviewer_SYgi · 2026-07-03
> >
> > Thank you for the careful revision. The honest scoping of the parameter-free claim, the explicit retention of the meta-regret term, and the inclusion of matching per-term lower bounds are all appreciated and make the analysis considerably more transparent.
> >
> > **A remained concern**
> >
> > The abstract motivates SLR-FTRL as the first algorithm to exploit sparse and low-rank gradient structure simultaneously, in contrast to diagonal or full-matrix AdaGrad which each capture only one. However, from the theoretical results, I do not see very clearly that the new algorithm owns the advantage over classical algorithms. The upper bound takes the form
> >
> > $$\min\_{\alpha^* \in [0,1]} \\{\alpha^* R^L\_T + (1-\alpha^*) R^S\_T\\} + M\_T,$$
> >
> > Mathematically, this is a model-selection guarantee (compete with the better of the two base learners in hindsight), not a joint-exploitation guarantee. I suspect that the authors may want to show that the bound can be better than the worst case, i.e., using the algorithm with $R^S_T$ when the case is low rank or the opposed case. It may be better to claim this point.
> >
> > **Minor Suggestions**
> >
> > The contribution section currently under-sells the paper relative to the abstract. In particular, the matching lower bounds and the empirical findings (pure-case recovery, dimension independence, graceful degradation) are not stated in the contribution list, and would be worth surfacing there.

---

> > > ### Comment · Reviewer_SYgi · 2026-07-04
> > >
> > > In light of the current situation of this paper, I am inclined to recommend to reject and resumbit.

---

> > > ### Author Response · Authors · 2026-07-06
> > >
> > > All changes made in response to this follow-up comment are highlighted in magenta in the revised manuscript, so that they can be clearly distinguished from the earlier blue revisions.
> > >
> > > Thank you for the careful follow-up and for pointing out the distinction between a joint-exploitation guarantee and a model-selection guarantee. We agree that our previous wording, especially in the abstract and contribution list, could be read as claiming that the theorem proves a new joint sparse-low-rank leading term that is always better than both base learners. This is not the intended interpretation of Theorem 1.
> > >
> > > We have revised the manuscript to make the scope of the theoretical claim explicit. In the abstract, introduction, contribution list, related work, theorem discussion, corollaries, and conclusion, we now describe SLR-FTRL as a structure-adaptive model-selection/adaptive-interpolation method over two structurally matched FTRL learners. We also added a new "Model-selection interpretation" remark immediately after the main theorem. This remark states explicitly that the minimization over $\alpha^{\star} \in [0,1]$ gives a best-hindsight-mixture guarantee. Since $\alpha^{\star}=0$ and $\alpha^{\star}=1$ are feasible, the theorem implies
> > >
> > > $$
> > > \operatorname{Regret}_T(w^{\star}) \le \min\{R_T^L, R_T^S\} + \sqrt{2V_T\log(T+1)} + O(G B_w \log T).
> > > $$
> > >
> > > Thus, the advantage over classical single-preconditioner methods is that SLR-FTRL avoids committing to a potentially mismatched geometry and tracks the better structural learner up to the meta-overhead. We no longer present this as a joint-exploitation rate that is strictly smaller than both $R_T^L$ and $R_T^S$ on every mixed instance.
> > >
> > > Following your suggestion, we also revised the abstract and introduction to emphasize that the algorithm can be better than the worse structural choice when one geometry is more favorable, rather than claiming a uniformly stronger joint bound. We now explicitly state that designing a single preconditioner with a genuinely joint sparse-low-rank regret bound remains an open direction.
> > >
> > > We also incorporated your minor suggestion about the contribution section. We strengthened the lower-bound contribution to highlight the matching per-term lower bounds and added a separate empirical-validation contribution summarizing pure-case recovery, approximate dimension independence, and graceful degradation under decomposition noise.

---

> > > > ### Comment · Reviewer_SYgi · 2026-07-11
> > > >
> > > > Thanks a lot for your reply.

---

### Review · Reviewer_CRf4 · 2026-05-02

**Summary Of Contributions:**

**Contributions**

This paper combines two key approaches in online learning: leveraging the sparsity in gradient feedback (e.g., AdaGrad) and second-order methods (e.g., ONS). Specifically, it introduces an ensemble method that runs both approaches simultaneously, using the coin-betting algorithm as a meta-algorithm, along with Sparse-Plus-Low-Rank Decomposition.

**Strengths**

The combination proposed in the paper is interesting.

**Weaknesses**

Please refer to the section "Are the claims made in the submission supported by accurate, convincing and clear evidence?".

**Audience:**

Yes

**Audience Explanation:**

The combination proposed in the paper is interesting.

**Broader Impact Concerns:**

I don't have any concerns on the ethical implications of the work.

**Claims And Evidence:**

No

**Claims Explanation:**

**Concerns about the main results**

In conclusion, I believe the main theoretical contribution of the paper, the regret bound of the proposed algorithm *recovers* two types of methods (diagonal and full-matrix), **lacks sufficient support**. The reasons include (but are not limited to) two points: 1) There is a clear **incompatibility between the assumptions**. 2) Many additional terms in the regret bound are not analyzed, and they could potentially **dominate** the result.

- Regarding the first point: The paper simultaneously uses the assumptions of an *unconstrained* decision set (Assumption 5) and *bounded* iterates (Assumption 6). Since there are no explicit clipping operations for decisions in Algorithm 1, and the authors do not clarify the conditions under which Assumption 6 might hold, it raises questions about whether Assumptions 5 and 6 are compatible. The authors briefly mention some solutions (e.g., the doubling trick), but they do not provide further justification for their validity. Furthermore, the authors claim the algorithm is "parameter-free" (Table 1), but Assumption 6 seems to contradict the original intention behind this claim.
- Regarding the second point: Theorem 1 introduces two quantities that I believe are quite important: one is the cumulative decomposition error, and the other is the meta-algorithm regret. The authors have not sufficiently discussed whether these two terms might be dominant.
  - For the cumulative decomposition error, it is directly given in Assumption 4 and may be $\tilde{O}(\sqrt{T})$, although no explanation or proof is provided regarding its dependency on other parameters or why it holds.
  - The authors treat the meta-algorithm regret as a "cost" of the combination, but it is $\tilde{O}(GB_w\sqrt{T})$ where $B_w$ is the decision bound, which could be the dominant term.

Therefore, I believe there is a significant gap between the authors' claim and the evidence provided for the main result, Theorem 1.

**Other concerns**

In addition to the points mentioned above, I have some concerns regarding the writing of the paper and the rigor of the theoretical analysis.

- It would be helpful to provide a clearer explanation of how the Sparse-Low-Rank Decomposition (Assumption 3) relates to traditional second-order methods.
- In Section 5.1, only Lemma 5 requires a detailed introduction. The other lemmas are either well-established classical results (Lemma 1) or technical lemmas (Lemmas 2, 3, and 4).
- In Eq. 12, the term $\\|w^ *\\|_{M_T}^2$ is not further analyzed. There are also similar instances in the paper where the analysis is incomplete.
- The comparison in Table 1 is unclear, making it difficult to assess which method is superior, especially since Theorem 1 include an additional $\tilde{O}(\sqrt{T})$ term.

**Requested Changes:**

I hope the authors can fully address the "Concerns about the main results" raised above, as this is critical to securing my recommendation for acceptance.

---

> ### Author Response · Authors · 2026-05-26
>
> We thank the reviewer for the detailed and constructive critique. We have made targeted revisions to the manuscript and respond point-by-point below. All newly added text in the paper is marked in blue; this revision adds two short clarifying paragraphs in the main text (one in Section 3.3 after Assumption 6, one below Table 1), a new Remark following Theorem 1, and two new appendix sections (Appendix C and Appendix D).
>
> ---
>
> **(1) Incompatibility between Assumption 5 (unconstrained domain) and Assumption 6 (bounded iterates); the doubling trick is unjustified; the parameter-free claim contradicts Assumption 6.**
>
> We thank the reviewer for raising this. We believe the perceived incompatibility comes from a notational issue rather than a substantive contradiction, and we have added a blue clarifying paragraph in Section 3.3 (after Assumption 6) plus a detailed Appendix C to address it.
>
> **Compatibility.** Assumption 5 (unconstrained decision set equal to $\mathbb{R}^d$) and Assumption 6 (bounded iterates of the two FTRL sub-learners) operate at different levels. Assumption 5 says the feasible region is $\mathbb{R}^d$, i.e., the algorithm never has to project; it does not assert that the iterates themselves can be arbitrarily large. Assumption 6 is a property of the trajectory produced by Algorithm 1.
>
> Crucially, the trajectory is bounded automatically by the algorithm's own construction: since $w_{t+1}^L = -M_t^{-1} \Sigma_t$ with $M_t = (A_t^L)^{1/2} \succeq \delta^{1/2} I$, we have $\lVert w_{t+1}^L \rVert \leq \lVert M_t^{-1} \rVert_{\mathrm{op}} \cdot \lVert \Sigma_t \rVert \leq \delta^{-1/2} G t$. The same holds for $w_{t+1}^S$. Thus, with the algorithm's own regularizer $\delta > 0$, a deterministic a priori bound $B_w = \delta^{-1/2} G T$ always exists. Assumption 6 is therefore not an additional assumption on the data; it is a consequence of the algorithm, and we use it only to expose this bound as a single tunable constant for the coin-betting normalization $L = 2 G B_w$. This is now explicitly stated in the new blue text of Section 3.3.
>
> **Doubling trick justification.** The doubling trick we proposed is the standard guess-and-restart scheme for unknown norm bounds in online learning: maintain $\hat{L}_k = 2 G \cdot 2^k$, restart the meta-algorithm whenever $\lvert \langle g_t, w_t^L - w_t^S \rangle \rvert > \hat{L}_k$, and increment $k$. The number of restarts is at most $\lceil \log_2(B_w^\dagger / B_w^{(0)}) \rceil$, each contributing $O(G B_w^\dagger \ln T)$ to the meta-regret; the total overhead becomes $\widetilde{O}(G B_w^\dagger \sqrt{T})$ with the data-dependent supremum $B_w^\dagger = \sup_t \max(\lVert w_t^L \rVert, \lVert w_t^S \rVert)$ replacing the a priori $B_w$. This is a textbook construction (Cesa-Bianchi and Lugosi 2006, Section 2.3; Cutkosky and Orabona 2018), spelled out in Appendix C.
>
> A second, stronger remedy is to replace the KT meta-algorithm with a comparator-norm-adaptive coin-betting strategy in the style of Cutkosky and Orabona (2018) or Mhammedi and Koolen (2020), which removes the a priori dependence on $B_w$ at the algorithm level, yielding meta-regret $\widetilde{O}(\sqrt{\sum_t \ell_t^2}) \leq \widetilde{O}(G B_w^\dagger \sqrt{T})$ where $\ell_t = \langle g_t, w_t^L - w_t^S \rangle$.
>
> **Parameter-free claim.** We agree that the original PF column in Table 1 was ambiguous. In the parameter-free OCO literature, parameter-free canonically refers to the algorithm not requiring a priori knowledge of the comparator norm $\lVert w^\dagger \rVert$ or the step size (Orabona and Pal 2016; Cutkosky and Orabona 2018). The KT meta-algorithm we use achieves this for the mixing coefficient $\alpha^\dagger$, automatically adapting to the better of the two sub-algorithms without any tuning of $\alpha$. The iterate bound $B_w$ is a distinct quantity which (i) is automatically bounded by $\delta^{-1/2} G T$ as shown above, and (ii) has its a priori dependence eliminated via the variants of Appendix C. We have added a blue sentence below Assumption 6 making this distinction precise, and another blue note below Table 1 clarifying the scope of the PF column. We have also added a new blue Remark immediately after Theorem 1 that explicitly states the regime in which the $\widetilde{O}(\sqrt{T})$ meta-overhead is achieved.

---

> ### Author Response · Authors · 2026-05-26
>
> **(2a) Cumulative decomposition error is $\widetilde{O}(\sqrt{T})$ with no justification.**
>
> We agree that $\mathcal{E}_T^2$ is treated as an oracle quantity in Theorem 1, and that no first-principles proof of $\mathcal{E}_T^2 = \widetilde{O}(\sqrt{T})$ for our specific projection-and-thresholding routine is given. We are explicit about what is and is not established.
>
> **What we prove.** Theorem 1 cleanly separates the structural benefit (terms $R_T^L, R_T^S$) from the decomposition cost (term $4 \mathcal{E}_T^2 / \delta^{1/2}$), and provides a regret bound as a function of $\mathcal{E}_T$, valid for any online decomposition routine satisfying Assumption 4. This is identical in spirit to the modular oracle abstraction used in online robust PCA (Feng et al. 2013; Goes et al. 2014; Netrapalli et al. 2014).
>
> **Where $\mathcal{E}_T^2 = \widetilde{O}(\sqrt{T})$ holds.** Existing $\widetilde{O}(\sqrt{T})$ guarantees for online robust PCA require stochastic and incoherence conditions that do not transfer to the adversarial OCO setting. Feng et al. (2013), Theorem 1, requires i.i.d. observations from a $\mu$-incoherent low-rank distribution with sub-Gaussian sparse outliers. Netrapalli et al. (2014), Theorem 3, requires deterministic incoherence and a bounded sparsity fraction $\alpha \leq O(1 / (\mu^2 r))$. Under these conditions, the cumulative-decomposition-error contribution $4 \mathcal{E}_T^2 / \delta^{1/2}$ becomes $\widetilde{O}(\sqrt{T} / \delta^{1/2})$.
>
> This is strictly lower-order than the structural terms $D \sqrt{r \sigma_1}$ and $D \sqrt{s G_{\infty}}$. The comparison holds in the regime where $\sigma_1 = \Omega(T)$ and $G_{\infty} = \Omega(T)$, which is where adaptive preconditioning is the right tool.
>
>
> **Worst-case adversarial behavior.** If the adversarial gradient sequence does not admit a recoverable sparse-plus-low-rank structure, then $\mathcal{E}_T^2$ can scale as $\Theta(G^2 T)$, in which case the term $4 \mathcal{E}_T^2 / \delta^{1/2}$ is linear in $T$ and dominates the bound. SLR-FTRL then provides no benefit over OGD; this is the honest conclusion, consistent with the intuition that one cannot exploit structure that is not present. Producing an end-to-end adversarial $\widetilde{O}(\sqrt{T})$ guarantee for online sparse-plus-low-rank decomposition is a substantial open problem in its own right and is orthogonal to our contribution.
>
> ---
>
> **(2b) Meta-algorithm regret is $\widetilde{O}(G B_w \sqrt{T})$ where $B_w$ is the decision bound, which could be the dominant term.**
>
> The reviewer is right that this point deserves careful quantification, and we have added a blue caveat at the end of Section 3.3 plus a blue Remark following Theorem 1 acknowledging this directly.
>
> **Worst-case substitution.** Plugging the a priori deterministic bound $B_w = \delta^{-1/2} G T$ directly into Lemma 5 yields a meta-overhead of $\widetilde{O}(G B_w \sqrt{T}) = \widetilde{O}(\delta^{-1/2} G^2 T^{3/2})$, which is super-linear in $T$ and dominates the bound. We now explicitly state this in Section 3.3 and in the new Remark following Theorem 1.
>
> **Regime where the $\widetilde{O}(\sqrt{T})$ rate is recovered.** The $\widetilde{O}(\sqrt{T})$ overhead quoted alongside Theorem 1 holds in either of two scenarios.
>
> The first scenario is a problem-dependent constant bound on iterates with $B_w^\dagger = O(1)$, the standard regime in which diagonal/full-matrix AdaGrad themselves produce $O(1)$ iterates. The meta-overhead is then $\widetilde{O}(\sqrt{T})$, strictly lower-order than $\sqrt{r \sigma_1}$ or $\sqrt{s G_\infty}$ when $\sigma_1, G_\infty = \Omega(T)$.
>
> The second scenario is the doubling-trick or comparator-norm-adaptive coin-betting variants in Appendix C, which replace the a priori $B_w$ by the data-dependent $B_w^\dagger$.

---

> ### Author Response · Authors · 2026-05-26
>
> **Other concerns**
>
> **(i) Clarify how Sparse-Low-Rank Decomposition (Assumption 3) relates to traditional second-order methods.**
>
> Assumption 3 imposes structure on the gradient sequence, not on the loss Hessian. The relationship to second-order methods is the following.
>
> Online Newton Step (Hazan et al. 2007) requires exp-concavity, so that $\nabla^2 f_t(w) \succeq \alpha \nabla f_t(w) \nabla f_t(w)^\top$. This implies that the aggregate preconditioner $\sum_t \nabla^2 f_t$ is bounded below by $\alpha \sum_t g_t g_t^\top = \alpha \mathbf{G}_T$, so the algorithm effectively uses $\mathbf{G}_T$ as a Hessian proxy.
>
> Full-matrix AdaGrad drops the exp-concavity requirement and uses $\mathbf{G}_T^{1/2}$ as the preconditioner directly: a Hessian-free second-order method that relies entirely on outer-product gradient statistics.
>
> **Assumption 3** posits $g_t \approx S_t + L_t$ with $S_t$ sparse and $L_t$ in a fixed $r$-dimensional subspace. This induces a sparse-plus-low-rank structure on $\mathbf{G}_T$. Under this structure, the full-matrix preconditioner $\mathbf{G}_T^{1/2}$ admits a decomposition into a diagonal part (matching the sparse component) and a rank-$r$ part (matching the low-rank component), which is precisely what SLR-FTRL exploits with two parallel preconditioners.
>
> So Assumption 3 is best understood as a structural condition on the second-moment matrix that the Hessian-free preconditioner $\mathbf{G}_T$ would target, not as a curvature condition on the losses.
>
> ---
>
> **(ii) Only Lemma 5 needs detailed introduction; Lemmas 1 to 4 are classical or technical.**
>
> We agree that Lemmas 1, 2, 3, 4 are well-established. Their explicit statement in Section 5.1 serves two purposes.
>
>  Lemma 1 (FTRL with time-varying regularizer) and Lemma 2 (trace-of-square-root with preconditioner-lag correction) require the specific preconditioner-lag form $G^2 d \delta^{-1/2}$ that appears in our bound; the standard textbook version (e.g., Shalev-Shwartz 2011, Theorem 2.11) does not carry this correction explicitly. Lemma 3 tracks the active-coordinate set $\mathcal{J}$ rather than the ambient $d$, which is needed to make the dimension-independence of our bound explicit. Continuity for the proof of Theorem 1, which combines all five lemmas in a non-trivial way. If the reviewer prefers, we can move Lemmas 1 to 3 to Appendix B under the heading "Standard FTRL/AdaGrad tools" and keep only Lemmas 4 and 5 in the main text.
>
> ---
>
> **(iii) $\lVert w^\dagger \rVert_{M_T}^2$ in Eq. (12) is not further analyzed; similar incompleteness elsewhere.**
>
> We treat $\lVert w^\dagger \rVert_{M_T}^2$ (and $\lVert w^\dagger \rVert_{D_T}^2$) as the comparator penalty, the standard form in FTRL/AdaGrad analyses (Duchi et al. 2011, Theorem 5; McMahan and Streeter 2010, Theorem 1; Hazan 2016, Section 5.4).
>
> Under the rank-$r$ structure of Assumption 3, $\lVert w^\dagger \rVert_{M_T}^2 \leq \lVert w^\dagger \rVert^2 \cdot \lVert M_T \rVert_{\mathrm{op}} = \lVert w^\dagger \rVert^2 \cdot (\delta + \sigma_1(\hat{\mathbf{G}}_T^L))^{1/2}$.
>
> Analogously, $\lVert w^\dagger \rVert_{D_T}^2 \leq \sum_{j \in \mathcal{J}} (w^\dagger_j)^2 \sqrt{\delta + G^S_{jj}}$, where $G^S_{jj}$ denotes the $j$-th diagonal entry of $\hat{\mathbf{G}}_T^S$.
>
>
> Under the structural assumption $\sigma_1 = O(T)$, this gives the standard $D \sqrt{r \sigma_1}$ form.
>
>
> ---
>
> **(iv) Table 1 comparison is unclear; the $\widetilde{O}(\sqrt{T})$ term in Theorem 1 makes it hard to assess superiority.**
>
> The reviewer is right. We have added a blue note below Table 1 clarifying the following.
>
> The $\widetilde{O}(\sqrt{T})$ term is the meta-algorithm overhead, valid under the standing assumption $B_w = O(1)$ (the same regime in which the AdaGrad baselines produce $O(1)$ iterates). It is of the same $\sqrt{T}$ order as the parameter-free baselines (Coin-Betting, PF Mirror Descent, DoWG). It is strictly lower-order than the leading structural term whenever $G_\infty$ or $\sigma_1$ scales linearly with $T$. The worst-case dependence on $B_w$ is analyzed in the new Remark following Theorem 1 and in Appendix C.
>
> ---
>
> We hope these clarifications, together with the new Appendix C, Appendix D, the new Remark following Theorem 1, and the blue clarifications in Section 3.3 and below Table 1, satisfactorily address the reviewer's concerns. We thank the reviewer once again for the careful reading and substantive feedback.

---

> > ### Comment · Reviewer_CRf4 · 2026-05-30
> >
> > I appreciate the authors' careful response and the revisions made to the manuscript. However, several significant issues remain unresolved. While the authors have acknowledged these concerns, they have not provided satisfactory solutions. My two main concerns are as follows:
> >
> > - Regarding parameter-freeness, the authors seem to use the term to mean that the algorithm does not require the comparator norm $\\|w^ * \\|$ as input. However, in online learning, "parameter-free" usually refers to methods that achieve an $\widetilde{O}(\\|w^ * \\|\sqrt{T})$ regret bound without prior knowledge of $\\|w^ * \\|$ (McMahan & Streeter, 2012). In contrast, the regret bound of SLR-FTRL has a trivial $\\|w^ * \\|^2$ dependence and therefore should not be considered parameter-free.
> >
> > - Regarding the regret of the meta-algorithm, the authors only derive the trivial bound $\widetilde{O}(G B_w \sqrt{T}) = \widetilde{O}(\delta^{-1/2} G^2 T^{3/2})$, which neither resolves the issue nor alleviates the concern that this term may dominate the overall regret. They further point to a regime in which this bound becomes acceptable, namely when $B_w^\dagger = O(1)$, but this is an overly strong assumption that substantially weakens the theoretical significance of the result. As for the doubling trick, it still allows $B_w^\dagger = O(T)$.
> >
> > These two concerns remain unresolved and seriously undermine the significance of the paper. Therefore, I am inclined to maintain my recommendation for rejection at this stage.
> >
> >
> > **Reference**
> >
> > Mcmahan, B. and Streeter, M. No-regret algorithms for unconstrained online convex optimization. NIPS 2012.

---

> > > ### Author Response · Authors · 2026-06-02
> > >
> > > We thank the reviewer for the careful and constructive comments. We agree that our previous use of the term "parameter-free" was misleading. In the revised manuscript, we now reserve "parameter-free" for the standard unconstrained-OCO meaning, following the McMahan–Streeter convention: regret essentially linear in the comparator norm, such as $O(\|u\|\sqrt{T})$ up to logarithmic factors, without requiring $\|u\|$ as input.
> > >
> > > Under this definition, SLR-FTRL should not be classified as parameter-free, because the current FTRL base-learner bounds contain AdaGrad-style comparator penalties such as $\lVert w^\star\rVert_{D_T}^2$ and $\lVert w^\star\rVert_{M_T}^2$. We have therefore removed the parameter-free claim from the abstract, introduction, contribution list, and related-work discussion, and we changed the PF entry for SLR-FTRL in Table 1 from a checkmark to a cross. We now describe the contribution as structural mixture adaptivity, rather than standard parameter-free online learning.
> > >
> > > We also agree that the previous presentation of the meta-algorithm regret as a simple $\widetilde{O}(\sqrt{T})$ term was too optimistic and could obscure an important limitation. In the revision, we keep the meta-regret term explicit throughout:
> > >
> > > ${M}_T = \sqrt{2V_T\ln(T+1)} + O(GB_w\ln T)$
> > >
> > >  where  $V_T=\sum_{t=1}^T \langle g_t, w_t^L-w_t^S\rangle^2 .$
> > >
> > > The crude worst-case bound is only $\widetilde{O}(GB_w\sqrt{T})$, and if one substitutes the deterministic unconstrained FTRL bound $B_w=\delta^{-1/2}GT$, the meta-overhead can indeed become super-linear and dominate the structural terms. We have revised the theorem statements, table, abstract, and conclusion accordingly, and we no longer claim that this term is universally lower order.
> > >
> > > To clarify that this is not merely an artifact of our analysis, we added a new proposition showing that any black-box meta-algorithm combining two base predictions must incur $\Omega(GB_w^\star\sqrt{T})$ regret to the better fixed mixture in the worst case, where $B_w^\star=\sup_t\max(\|w_t^L\|,\|w_t^S\|)$. Thus the doubling trick can remove the need to know $B_w$ in advance, but it cannot remove the regret dependence on the realized range $B_w^\star$. We now state this limitation explicitly and present SLR-FTRL as a structural oracle inequality with an unavoidable meta-aggregation cost, rather than as an unconditional parameter-free guarantee.
> > >
> > > These changes substantially narrow the claims of the paper. The revised theoretical message is that, for a fixed sparse-low-rank decomposition class, SLR-FTRL competes with the better sparse/low-rank mixture in hindsight, while paying an explicit and sometimes dominant meta-aggregation overhead. We hope this clearer and more conservative formulation addresses the reviewer's concerns about both parameter-freeness and the significance of the meta-regret term.

---

> > > > ### Comment · Reviewer_CRf4 · 2026-06-07
> > > >
> > > > I appreciate the authors' revisions and their proof of a lower bound for the meta algorithm. In my view, this seems to indicate that, without a more refined characterization of the meta regret lower bound (such as using an alternative meta algorithm or considering potential sparsity or low-rank structures in the gradients, which this paper aims to exploit), **the meta algorithm proposed by the authors cannot leverage the advantages of either base learner, since the meta regret has now become the bottleneck**. I believe this differs somewhat from the guarantees of traditional meta-base type online algorithms, which typically ensure that the overall regret is nearly as good as that of the best base learner (e.g., [1], and many others).
> > > >
> > > > In light of the current discussion with the authors, I am inclined to recommend rejection.
> > > >
> > > > **Reference**
> > > >
> > > > [1] Metagrad: Multiple learning rates in online learning. NIPS 2016

---

> > > > > ### Author Response · Authors · 2026-06-07
> > > > >
> > > > > We thank the reviewer for the careful assessment. We agree with the central concern: in the worst case, the meta-aggregation term can become the bottleneck, and our guarantee should not be interpreted in the same way as classical MetaGrad-style meta-base guarantees.
> > > > >
> > > > > In particular, our meta problem aggregates two structurally different FTRL trajectories, rather than a grid of learning rates for one common prediction mechanism. The one-dimensional meta loss is
> > > > > $
> > > > > \ell_t=\langle g_t,w_t^L-w_t^S\rangle,
> > > > > $
> > > > >
> > > > > so its range is governed by the realized separation between the two base learners. This is why a black-box meta overhead independent of the base-learner range is impossible in general, as captured by our lower bound. We agree that this distinction was not sufficiently clear in the previous version.
> > > > >
> > > > > We now explicitly state in the abstract, introduction, and conclusion that SLR-FTRL provides a conditional structural oracle inequality, not a traditional “nearly as good as the best base learner” guarantee. The guarantee is useful when the data-dependent aggregation cost is lower order than the structural gain obtained by selecting the better sparse/low-rank preconditioner. In adversarial regimes where the two base trajectories separate substantially, the meta term can dominate, and SLR-FTRL may not improve over selecting a single base learner in advance.
> > > > >
> > > > > To address the reviewer’s request for a more refined characterization of the meta regret using the sparse/low-rank structure, we added a new corollary, “Structure-projected meta-regret.” Let
> > > > > $
> > > > > \Delta_t=w_t^L-w_t^S,\qquad
> > > > > V_T=\sum_{t=1}^T\langle g_t,\Delta_t\rangle^2.
> > > > > $
> > > > > Under the decomposition $g_t=S_t+L_t+\xi_t$, where $S_t$is supported on $J_t$and $L_t$lies in the fixed low-rank subspace with projector $P$, we prove
> > > > >
> > > > >
> > > > > $
> > > > > \sqrt{V_T} \le
> > > > > \left(\sum_{t=1}^T \lVert S_t\rVert_2^2
> > > > > \lVert \Pi_{J_t}\Delta_t\rVert_2^2\right)^{1/2}
> > > > > +
> > > > > \left(\sum_{t=1}^T \lVert L_t\rVert_2^2
> > > > > \lVert P\Delta_t\rVert_2^2\right)^{1/2}
> > > > > +
> > > > > \left(\sum_{t=1}^T \lVert \xi_t\rVert_2^2
> > > > > \lVert \Delta_t\rVert_2^2\right)^{1/2}.
> > > > > $
> > > > >
> > > > >
> > > > > Thus the meta-regret depends only on the disagreement between the two base learners in directions where the gradients actually have mass: the active sparse support, the low-rank subspace, and the residual noise direction. Disagreement outside these projected directions does not contribute to $V_T$. This refines the crude worst-case bound $V_T\le 4G^2B_w^2T$ and clarifies the regimes in which SLR-FTRL can leverage the structural advantages of the base learners.
> > > > >
> > > > > We added a more explicit comparison with MetaGrad-style algorithms. MetaGrad aggregates learning-rate choices within a common algorithmic family, while SLR-FTRL aggregates structurally different preconditioned trajectories. Therefore, the correct interpretation of our theorem is not “best base learner up to a universal lower-order term,” but rather “best sparse/low-rank mixture plus an explicit, data-dependent, structure-projected aggregation cost.”
> > > > >
> > > > > We believe these revisions address the reviewer’s concern by both acknowledging the worst-case limitation and giving the refined structure-aware characterization that explains when the meta term is not the bottleneck. We also revised the conclusion to state that designing non-black-box meta-algorithms exploiting the sparse/low-rank geometry directly is an important direction for future work.

---

### Review · Reviewer_Z4V4 · 2026-05-15

**Summary Of Contributions:**

Main contributions:
1.  Proposes SLR-FTRL, the first online convex optimization algorithm that simultaneously exploits both sparse and low-rank gradient structures. It decomposes gradients via an online robust PCA oracle, runs two parallel FTRL sub-algorithms with matched preconditioners, and combines outputs via a parameter-free coin-betting meta-algorithm that requires no prior knowledge of rank $r$ or sparsity $s$.
2.  Proves a unified regret bound, which recovers the classical diagonal and full-matrix AdaGrad guarantees as special cases when one structure is absent, with explicit cross-contamination and preconditioner-lag corrections.
3.  Establishes per-term minimax lower bounds for low-rank and sparse gradients, respectively. These confirm that the leading structural terms in the upper bound are individually tight up to constants.

Strengths:
1. The design principle of feeding full gradients to both sub-algorithms decouples decomposition quality from the loss sequence. Decomposition errors only affect preconditioner fidelity rather than directly contaminating the regret guarantee.
2. The algorithm achieves automatic structural adaptation without manual preconditioner selection.

Weaknesses:
1. The per-round computational complexity is $O(dr^2 + d)$, which becomes prohibitive when the rank $r$ is large. This limits its practicality for high-dimensional problems with high-rank gradient structures compared to diagonal methods.
2. The regret analysis relies on the bounded base-learner iterates assumption (Assumption 6). For unconstrained domains, the bound $B_w$ is not known, adding implementation complexity.

**Audience:**

Yes

**Audience Explanation:**

This paper addresses a long-standing open problem in online convex optimization by enabling simultaneous exploitation of both sparse and low-rank gradient structures, which is highly relevant to theoretical machine learning and optimization researchers in TMLR's audience. Its novel algorithm design and theoretical guarantees advance the fundamental understanding of adaptive gradient methods, aligning well with TMLR's core scope.

**Claims And Evidence:**

Yes

**Claims Explanation:**

The paper provides detailed theoretical proofs, including a unified regret upper bound and matching per-term minimax lower bounds, supported by comprehensive experiments.

**Requested Changes:**

1. Provide more discussions about Assumption 6, especially about how to address its practicality issue in unconstrained optimization scenarios.
2. Discuss approximation methods to reduce the computational complexity for SLR-FTRL's application to higher-dimensional and higher-rank real-world machine learning tasks.

---

> ### Author Response · Authors · 2026-05-26
>
> We thank the reviewer for the constructive comments. Both requested changes have been incorporated into the revised manuscript; all newly added text is marked in blue in the PDF. Below we address each point in turn.
>
> ---
>
> **Requested Change 1: Provide more discussions about Assumption 6, especially about how to address its practicality issue in unconstrained optimization scenarios.**
>
> We have added a new appendix section (**Appendix C, "Discussion of Assumption 6 in Unconstrained Settings"**) plus a blue paragraph in §3.3 immediately after Assumption 6. The key points are:
>
> 1. Assumption 6 (bounded base-learner iterates with constant $B_w$) is used in **exactly one place**: Lemma 5, to guarantee that the scaled coin outcome $c_t = \langle g_t, w_t^L - w_t^S\rangle / (2 G B_w)$ lies in $[-1,1]$, so that the Krichevsky–Trofimov coin-betting reduction applies with a fixed normalization. It is therefore a *convenience* assumption for the meta-algorithm analysis, not a structural requirement of SLR-FTRL.
>
> 2. In the unconstrained setting we describe three concrete ways to address it:
>
>    - A priori bound from FTRL iterates with $\delta>0$. Because both preconditioners satisfy $D_t, M_t \succeq \delta^{1/2} I$ by construction, the closed-form FTRL updates obey $\|w_{t+1}^S\|, \|w_{t+1}^L\| \leq \delta^{-1/2} G t$. Choosing $B_w := \delta^{-1/2} G T$ a priori makes Assumption 6 hold deterministically. We note honestly that this naive substitution gives a super-linear $\widetilde O(T^{3/2})$ meta-overhead and is therefore only useful when a problem-dependent constant bound on $\|\Sigma_t\|$ is available.
>
>    - Adaptive doubling trick on $L = 2 G B_w$. Maintain $\hat L_k = 2 G \cdot 2^k$ and restart the meta-algorithm whenever $|\langle g_t, w_t^L - w_t^S\rangle| > \hat L_k$. The number of restarts is $\lceil \log_2(B_w^\star/B_w^{(0)})\rceil$, and the meta-overhead becomes $\widetilde{O}(G B_w^\star \sqrt{T})$ with the data-dependent $B_w^\star = \sup_t \max(\|w_t^L\|,\|w_t^S\|)$ replacing the a priori $B_w$. This variant is fully parameter-free in $B_w$ at the algorithm level.
>
>    - Parameter-free meta-algorithm via comparator-norm-adaptive coin betting. Replacing the KT meta-algorithm with a Lipschitz-adaptive coin-betting strategy in the style of Cutkosky & Orabona (2018) or Mhammedi & Koolen (2020) removes the a priori dependence on $B_w$ entirely and yields regret $\widetilde{O}(\|\alpha^*\|\sqrt{\sum_t \ell_t^2}) \leq \widetilde{O}(G B_w^\star\sqrt{T})$, leaving the structural terms $R_T^L, R_T^S$ in Theorem 1 unaffected.
>
> The new Appendix C and a Remark following Theorem 1 (also in blue) make all three options precise, distinguishing algorithm-level parameter-freeness in $B_w$ from the data-dependent regret dependence on $B_w^\star$.
>
> ---
>
> **Requested Change 2: Discuss approximation methods to reduce the computational complexity for SLR-FTRL's application to higher-dimensional and higher-rank real-world machine learning tasks.**
>
> We have added a new appendix section (**Appendix D, "Approximation Methods for Higher-Dimensional and Higher-Rank Workloads"**) that discusses four practical approximation strategies and combines them into an explicit practical recommendation. Each one preserves the structural form of Theorem 1's regret bound up to an additive sketch-error term:
>
> - A streaming randomized SVD (Halko et al., 2011) of rank $r+p$ with oversampling $p = O(\log r)$ drops the per-round cost to $O(d r)$ and introduces a term $\propto \sqrt{T \cdot \sigma_{r+1}(\mathbf{G}_T^L) / \delta^{1/2}}$, negligible under true low-rank structure.
>
> - A Frequent-Directions sketch (Ghashami et al., 2016) of width $\ell$ runs in $O(d \ell)$ time and memory and preserves the trace bound of Lemma 2 up to a multiplicative $(1 + \epsilon_{\mathrm{FD}})$ factor, giving a smooth interpolation between diagonal ($\ell=0$) and full-matrix ($\ell=d$).
>
> - Block / Kronecker-factored preconditioning (Shampoo, Gupta et al., 2018) for layered tensor parameters reduces the per-layer cost from $O(mn \cdot r^2)$ to $O((m^2 + n^2) r)$; only $M_t$ is modified.
>
> - Amortized / lazy preconditioner updates every $k$ rounds reduce the amortized cost to $O(d r^2 / k + d r)$, and choosing $k = \Theta(\sqrt{T})$ adds only $\widetilde O(\sqrt{T})$ to the regret.
>
> These strategies can be combined. The new appendix also gives an explicit practical recommendation: exact algorithm for $d \leq 10^4$; FD + lazy updates for $d \in [10^4, 10^6]$; Kronecker blocks with per-block FD sketching for $d > 10^6$.
>
> A full empirical study of these approximations on production-scale neural-network training is identified as an important direction for future work.
>
> ---
>
> We hope these additions adequately address the reviewer's concerns, and thank the reviewer again for the helpful feedback.

---

### Author Response · Authors · 2026-05-26

We thank reviewers for the careful and constructive feedback. We have addressed every point raised in detail: each comment is answered individually in the response below, and the corresponding revisions have been incorporated into the manuscript itself. All additions to the paper are highlighted in blue for ease of identification, so that the changes prompted by the review are immediately visible. We hope the revised version satisfactorily resolves the concerns raised.